



# Complex interactions of in-stream DOM and nutrient spiralling unravelled by Bayesian regression analysis

Matthias Pucher[1,2], Peter Flödl[3], Daniel Graeber[4], Klaus Felsenstein[5], Thomas Hein[1,2] and Gabriele Weigelhofer[1,2]

[1] WasserClusterLunz – Biologische Station GmbH, Lunz am See, Austria
[2] Institute of Hydrobiology and Aquatic Ecosystem Management, University of Natural Resources and Life Sciences, Vienna, Austria
[3] Institute of Hydraulic Engineering and River Research, University of Natural Resources and Life Sciences, Vienna, Austria
[4] Department Aquatic Ecosystem Analysis and Management (ASAM), Helmholtz Centre for Environmental Research – UFZ, Magdeburg, Germany
[5] Department of Statistics, Vienna University of Technology, Vienna, Austria

*Correspondence to*: Matthias Pucher (matthias.pucher@wcl.ac.at)

**Abstract.** Uptake and release patterns of dissolved organic matter (DOM) compounds and nutrients are entangled, and the current literature does not provide a consistent picture of the link between DOM composition, nutrient concentrations, and effects on their cycling. We performed two plateau addition experiments for each of five different, realistic, complex DOM leachates in a small stream, heavily enriched in nitrate but not phosphate or DOM due to diffuse agricultural pollution. By including cow and pig dung as well as corn, leaves and nettles leachates, the study used a wide range of different DOM qualities. We measured changes in nutrient concentrations and determined DOM fractions by fluorescence measurements and parallel factor (PARAFAC) decomposition. To assess influences from hydrological transport processes, we used a 1-D hydrodynamic model.

We propose a non-linear Bayesian approach to the nutrient spiralling concept, the Interactions in Nutrient Spirals using BayesIan REgression (INSBIRE) approach. This approach can disentangle complex and interacting biotic and abiotic drivers in nutrient uptake metrics, show their variability and quantify their error distribution. Furthermore, previous knowledge on nutrient spiralling can be included in the model using prior probability distributions. We used INSBIRE to assess interactions of compound-specific DOM and nutrient spiralling metrics the data of our experiment.

The uptake processes of different DOM fractions were linked to each other. We observed stimulating and dampening effects of DOM fractions on each other and the overall DOM uptake. We found saturation effects for dissolved organic carbon (concentration of C, DOC) uptake, as rising concentrations of a DOM fraction dampened its uptake. The degradation of a humic DOM component of terrestrial origin was stimulated by other DOM fractions, pointing to priming effects. We also found an influence of the wetted width on the uptake of soluble reactive phosphorus (SRP) and a microbially derived humic substance, which indicates the importance of the sediment-water interface for P and humic C cycling in the studied stream. Interestingly, we found no interactions between DOM uptake and nitrate or SRP concentrations, or any effect of the added



DOM leachates on nitrate uptake, indicating that the increase in DOC concentrations and SRP concentrations were not sufficient to affect the relatively steady nitrate uptake during the experiments.

Overall, we show that bulk DOC is a weak predictor of DOC uptake behaviour for complex DOM leachates and that individual DOM compound uptake, nitrate uptake and SRP uptake are controlled very differently within the same aquatic ecosystem. We also found effects of hydromorphology on the uptake of one humic fluorophore and SRP. We conclude that cycling of different C fractions, their interaction and interactions with N and P uptake in streams is a complex, non-linear problem, which can only be assessed with advanced non-linear approaches, such as we present with INSBIRE.

## 40    1 Introduction

Dissolved organic matter (DOM) in freshwater ecosystems is an important part of the global carbon cycle (Battin et al., 2009; Cole et al., 2007; Creed et al., 2018). It strongly influences various biogeochemical processes. Quantity and quality of DOM relate to respiration in streams (Niño-García et al., 2016), rivers (Besemer et al., 2009), and estuaries (Amaral et al., 2016). DOM also controls bacterial activity and influences the bacterial community composition (Freixa et al., 2016).

Furthermore, DOM can modify nitrate ($N-NO_3$) uptake (Taylor and Townsend, 2010; Wymore et al., 2016(Taylor and Townsend, 2010; Wymore et al., 2016)) and influence the toxicity of pesticides (Bejarano et al., 2005(Bejarano et al., 2005) (Bejarano et al., 2005)).

Streams can retain dissolved nutrients and organic matter imported from the terrestrial catchment (Weigelhofer et al., 2018b). This capacity provides the basis for good water quality in receiving water bodies (Ensign and Doyle, 2005).

Environmental factors and human impacts within the watershed influence both the transport of terrestrial DOM to streams and the in-stream processing (Battin et al., 2008; Giling et al., 2014; Graeber et al., 2012, 2015; Hedin et al., 1995; Manzoni and Porporato, 2011; Mattsson et al., 2009; Wilson and Xenopoulos, 2009). Agriculture, for example, has been shown to change the amount and composition of the DOM in stream ecosystems as well as the related microbial communities (Eder et al., 2015; Findlay et al., 2001; Findlay and Sinsabaugh, 2003; Graeber et al., 2012). However, the effects of changed DOM

and nutrient supply on the DOM and nutrient uptake in streams remains in the dark.

In-stream DOM uptake and retention is mostly related to the stoichiometry of the organic carbon supply (i.e. the ratio of dissolved organic carbon (C) to dissolved nitrogen (N) and phosphorus (P), Graeber et al., 2015; Gücker et al., 2016) as well as to the structure and the bioavailability of the individual DOM compounds (Mineau et al., 2016). While a considerable part of the reactive N and P is bound in small and simple molecules, dissolved organic C is bound in a mixture of differently

structured organic molecules, whose retention times vary by several orders of magnitude. The dissolved organic carbon (concentration of C, DOC) uptake processes are more difficult to assess, because a variety of new compounds is produced during decomposition (Nebbioso and Piccolo, 2011). These changes in composition explain why quality-related mass balance approaches (e.g. Schiller et al., 2011) are futile without knowing the exact transformation pathways. However, DOM and nutrient retention capacities can also be studied by measuring the net retention of an artificially increased concentration



between longitudinal sampling points in a stream (Mineau et al., 2013; Stream Solute Workshop, 1990; Weigelhofer, 2017; Weigelhofer et al., 2018b). We expect a complex interaction between the different DOM fractions and the available N and P to explicate the bioavailability and the aquatic retention of the DOM. However, these interactions are difficult to quantify. This study aims to provide a first approach to quantify complex DOM, N and P interactions and their combined role in the overall DOM and nutrient retention in an agricultural stream impacted by diffuse nutrient pollution.

Our field experiment comprised several in-stream short-term plateau additions with different DOM sources in an agriculturally influenced headwater stream according to the nutrient spiralling concept (Stream Solute Workshop, 1990). Because of the diverse composition of DOM, we needed a way to analyse interactions between different DOM components and nutrients, including uncertainty propagation. Therefore, we decided to use a Bayesian approach, because it is a suitable tool for ecological and biogeochemical questions, allowing us to assess natural variability, and assign degrees of belief in

hypotheses based on measured data (Arhonditsis et al., 2008; Berger and Berry, 1988; Cox, 1946; Ellison, 2004; Jaynes, 2003; McCarthy, 2007). We used data from previous studies (e.g. Mineau et al., 2016) and expert knowledge to define prior distributions for the used parameters. We derived posterior distributions of the uptake parameters rather than single values. Previous studies have observed and modelled nutrient efficiency loss in uptake processes (Dodds et al., 2002; O'Brien et al., 2007). The efficiency loss model describes a non-linear increase of uptake rates with increasing concentrations following a

power function with an exponent lower than 1. The dampening effects of nutrient concentration on the uptake efficiency can be extended to quantify stimulating effects in retention as well and can be included in the nutrient spiralling equations. By that, the parameters are calculated from the measured values directly and measurement errors can be compared with model errors in an uncertainty propagation analysis. We reached our requirements on the data analysis by (1) adding dampening and stimulating effects, comparable to nutrient efficiency loss, to the nutrient spiralling equations, (2) restructuring these

equations to solve them in one step and (3) using a Bayesian algorithm to fit the parameters. We called this approach Interactions in Nutrient Spirals using BayesIan REgression (INSBIRE). With INSBIRE, we addressed the following questions:

1. What are the differences in bulk DOC uptake velocity of different leachates?
2. How do selected DOM components behave in comparison to the bulk DOC uptake velocity?

3. Which factors and interactions influence the uptake velocity of the bulk DOC as well as the uptake of the individual DOM components and the co-transported nutrients N and P?

## 2 Methods

### 2.1 Site description

The experiment was carried out in the Hydrological Open Air Laboratory (HOAL, Figure 1) in Petzenkirchen, Austria

(Blöschl et al., 2016). The 1st order stream has several inflows, two natural springs, six drainage pipes, and one site with





groundwater infiltration from a small wetland. The stream is characterized in sections by (dense) grass growth on the banks, with deciduous forest dominating at the beginning and end of the study reach. All inflows as well as the stream discharge are continuously monitored regarding water quantity and quality. Sediments are dominated by clay washed in from the adjacent fields during storm events. Table 1 shows the extent and basic environmental characteristics of the stream.

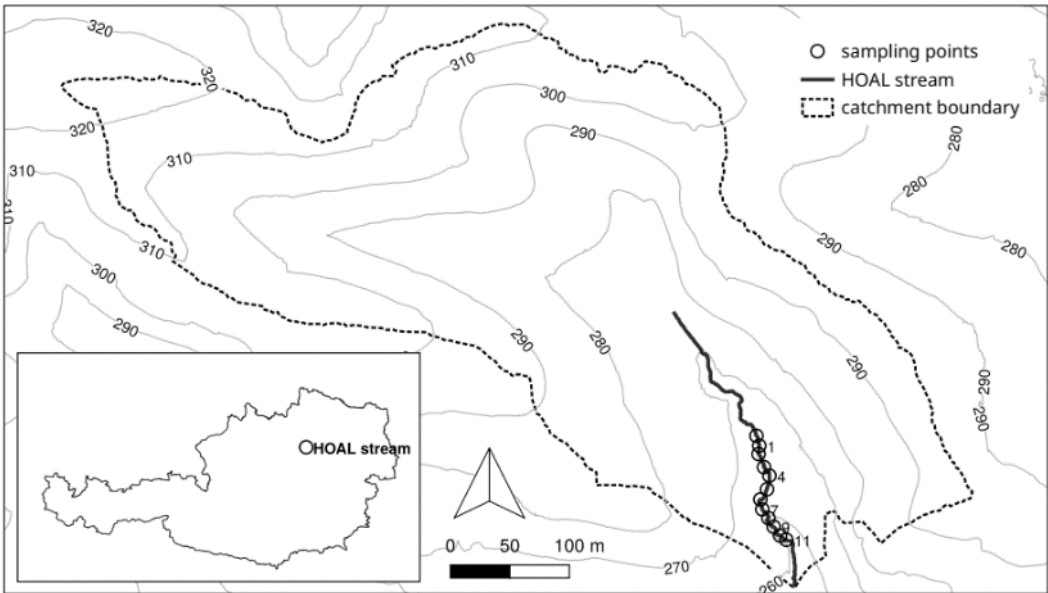


**Figure 1: Hydrologic open-air lab HOAL: catchment, stream, sampling points and location within Austria**

**Table 1: Extent and environmental characteristics of the HOAL**

| Characteristic | Value | Unit |
|---|---|---|
| Length | 620 | m |
| Catchment size | 0.66 | km$^2$ |
| Arable land coverage | 90 | % |
| Mean annual discharge | 0.004 | m$^3$s$^{-1}$ |
| Peak discharge | 2 | m$^3$s$^{-1}$ |
| Mean annual temperature | 9.5 | °C |
| Mean annual precipitation | 820 | mm yr$^{-1}$ |

To avoid any lateral inflow, we chose a reach of 215 m situated between two lateral inflows for the experiments. We divided

the study site into subsections of 16 to 26 m, depending on accessibility. The stream is characterized by a meandering course but is stretched with frequent pools (up to 24 cm in depth) at the end of the study reach. Between point 4 and point 5, *Equisetum palustre* and *Juncus sp.* grow in this open section's water (Figure 2). At point 7, the patchy canopy cover



facilitates the growth of algae on the stream bed. During the experiment, the median temperature was 16.7 °C (IQR = 2.4) and the median conductivity was 633 µS cm$^{-1}$ (IQR = 23).

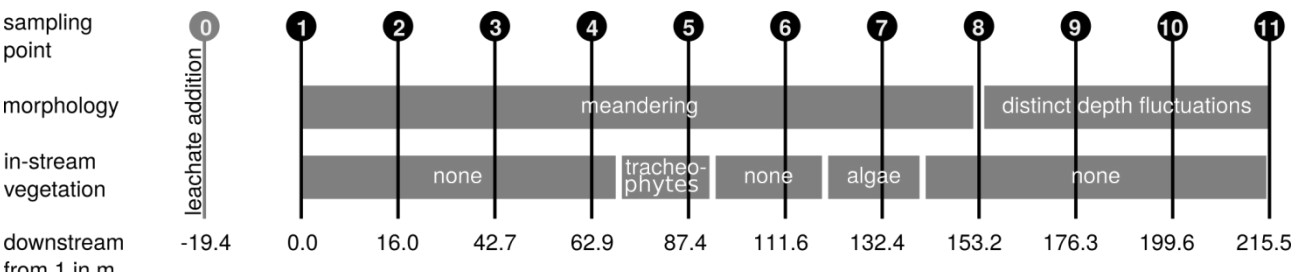

**Figure 2: Sampling scheme and general parameters of the stream.**

## 2.2 Experimental design

The experiment was performed during six consecutive weeks in July and August 2018. No major rain event occurred in the study area and the average discharge was between 0.38 and 0.93 l s$^{-1}$. Ten additions with DOM leachates from five different sources were injected into the study reach using short-term plateau additions according to the Stream Solute workshop protocol (Stream Solute Workshop, 1990; Weigelhofer et al., 2012). The respective leachate plus a NaCl solution as conservative tracer were pumped into the stream over 2 to 2.5 hours via a peristaltic pump. We used a mobile conductivity meter to identify plateau conditions in the stream at each sampling point. During plateau conditions, water samples were taken at each sampling point for the analysis of nutrient concentrations, organic carbon concentrations, and DOM composition. By that, we followed one virtual water package travelling downstream and took samples at different points in time. The leachates were introduced at point 0. The first sampling point was chosen to ensure full mixing with the stream water. After shutting the addition off, the change in conductivity was recorded until salt concentrations had returned to ambient levels. We added leachates one or two times per week, at least one day apart. Similar leachates were used five to seven days apart to prevent adaption of the microbial community and interferences among leachates. Each Monday, we sampled during ambient concentrations to interpolate background conditions for the days with addition experiments. All samples were taken between 10:00 and 14:00 to ensure comparability.

## 2.3 Preparation of the leachates

The leachates were prepared from 50 g l$^{-1}$ dry matter of cow and pig dung, foliage from local trees (*Acer platanoides, Acer pseudoplatanus, Lonicera xylosteum, Pteridium aquilinum, Sambucus nigra*), nettles (*Urtica dioica*), and corn plant (*Zea mays*) leaves. We leached with nutrient-poor water from a local well under aerated conditions in a barrel over 24 h. The leachates were filtered in steps of 2 mm and 0.5 mm using stainless steel sieves and 50 µm using a 25 cm spun filter cartridge (PureOne PS-10). The end volume was between 40 and 60 l. To avoid post-leaching changes in DOM, the leachates were prepared freshly for each addition.

Average DOC concentrations in the stream water were about 1.3 mg l$^{-1}$. We aimed to achieve an increase by about 3 mg l$^{-1}$
DOC in the experiments. Some sources proved difficult to leach in sufficient amounts and parts of the leached DOC was
degraded even during short storage. Thus, the DOC increase achieved during the experiments was between 0.2 and
2.3 mg l$^{-1}$. Even within the same source, leached amounts varied in concentration and composition between different
additions. We consider this unproblematic since we defined the leachates by their measured composition and not solely by
their source. On the contrary, the fluctuations broaden the distributions of measured values and can provide more stable
models as well as a more general picture of the uptake processes.

**2.4 Analyses**

Before the analyses in the lab, samples were filtered through combusted Whatman glass microfiber filters, Grade GF/F (0.7
µm) for syringes. We measured inorganic nitrogen as N-NO$_3^-$, nitrite (N-NO$_2^-$) and ammonium (N-NH$_4^+$)as well as soluble
reactive phosphorus (SRP) with a Continuous Flow Analyzer (accuracy ± 0.1 µg l$^{-1}$). Dissolved organic carbon (DOC) was
measured with a Sievers*900 portable TOC-Analyzer (accuracy ± 2%). We measured the DOM quality (Excitation-
Emission-Matrices) via Fluorescence Spectroscopy with a Hitachi Fluorescence Spectro-photometer F-7000 and DOM
absorbance with a Shimadzu UV-1700 spectrophotometer.
We analysed the data using R software version 3.5 (R Development Core Team, 2019) and tidyverse (Wickham et al., 2019).
The DOM EEMs were pre-processed using eemR (Massicotte, 2019), the PARAFAC analysis was done with staRdom
(Pucher et al., 2019). The measured fluorescence EEMs were corrected for inner-filter effects, samples of ultra-pure water
were subtracted, scatter bands were removed and interpolated and the samples were normalized to Raman units. Samples
were screened visually and no unusual noise was found. After obtaining first models, three outliers were identified using the
samples' leverages and excluded from the model. The components' spectra were visually checked for plausibility. After that,
a suitable model was validated using a split-half analysis. The final model did not express any problems related to those
criteria. The outliers were included again to calculate loadings under the already fixed components. For calculating the
PARAFAC models and the split-half validation, we used 256 random initializations, a tolerance of 10$^{-11}$ and staRdom's
standard way to split the data.

**2.5 Hydrodynamic modelling**

A hydrodynamic 1D-model was used to calculate the necessary hydraulic parameters using the software package HEC-RAS.
For the creation of the terrain model, a cross-sectional approach was applied, where 64 cross-sections were recorded at a
distance of 0.8 m to 6.8 m depending on structural variations and accessibility. A total of 251 points were measured in the
stream with a theodolite (Leica TC805) and then merged with a 1 x 1 m floodplain area model (based on the official laser
scan data of the province of Lower Austria) using the software package Surface-water Modeling System (Aquaveo,
LLC).The model was calibrated with the discharge data recorded at the HOAL site by comparing the measured water surface





elevation with the modelled one. The calibrated 1D model was used to calculate the hydraulic parameters flow velocity, water depth and wetted width at each sampling point for each sampling day.

**2.6 Bayesian non-linear regression**

The nutrient uptake was calculated using a Bayesian non-linear model and solved with a Markov chain Monte Carlo (MCMC) algorithm as provided in the R package brms (Bürkner, 2017) relying on stan (Carpenter et al., 2017). The basic

principle behind MCMC is to alternately sample parameter values from given prior distributions and determine the model's goodness of fit resulting in a posterior distribution for each parameter. These distributions show plausible ranges, stemming from measurement errors, variability in nature and not modelled influences for each parameter.

For model comparisons, we used the Bayes factor (BF, Goodman, 1999a, 1999b), which is the likelihood ratio of the marginal likelihood of two competing models. A Bayes factor of 10 in favour of a particular model means that this model is

10 times more likely to explain the measured data. The interpretation of the Bayes factor was conducted according to (Kass and Raftery, 1995). In that way, a Bayes factor of more than 3.2 is considered to show substantial evidence, while values below are barely noteworthy. A BF <1 corresponds to the inverse of the BF, but in favour of the other hypothesis. Selecting models with the Bayes factor also allows removing models prone to collinearity problems (Ghosh and Ghattas, 2015). The Bayes $R^2$ (Gelman et al., 2019) for each model was calculated to relate our results to this commonly used parameter and

demonstrate the accuracy of the analysis. It was not used for performance measurements.

**2.7 Calculating Interactions in Nutrient Spirals using BayesIan REgression (INSBIRE)**

We used the equations (Eqs. 1-3 and 5 below) of the nutrient spiralling concept provided by the Stream Solute Workshop (1990) to develop our solute spiralling model INSBIRE. For a straightforward solving scheme, a single-step analysis is necessary to determine the posterior distributions of all interdependent parameters at once. Interactions, model weaknesses,

collinearity (Ghosh and Ghattas, 2015), and the variation of parameters can then be assessed and interpreted in a consistent way. Values along the stream were measured in a longitudinal series which is formally identical to a time series problem. We re-arranged the equations so that differences are replaced by current (e.g. $C_x$) and past (e.g. $C_{x-1}$) values of series. These equations conform to a time series including past values of the same variable as well as current and past values of other variables and are a form of non-linear autoregressive exogenous models (NARX, e.g. Billings, 2013). Several studies used

the original equations of the Stream Solute Workshop protocol (1990) and solved them via variable transformation. Still, the results from a linear regression using transformed data and those of a direct non-linear fit differ (e.g. Stedmon et al., 2000). Therefore, we regard a non-linear solving algorithm superior in terms of accuracy.

In order to compare models of similar shape, we proceeded differently, transforming all equations into a NARX form, which yields Eq. (8). Commonly, uptake length ($s_w$), uptake velocity ($v_f$) and areal uptake rate (U) are used in nutrient uptake

studies (Dodds et al., 2002; O'Brien et al., 2007; Trentman et al., 2015; Weigelhofer et al., 2018b). We used all three approaches and fitted our experimental results to the Eqs (2), (4) and (6). $s_w$ is known to change with different discharges,




while $v_f$ should compensate this problem (Dodds et al., 2002). We choose priors to approximately fit knowledge from other studies (e.g. Mineau et al., 2016) while keeping them broad, so they do not dominate the results. Priors and especially their limits were adjusted to deliver converging models. We provide an exemplary R script that demonstrates INSBIRE (Pucher, 2020).

We used data from all experiments combined to fit Eqs. (2), (4), (6) and (8). By that, we increased the number of points to fit a model which enabled us to get more general insight into processes and estimate interactions that can only be observed with different nutrient and DOM ratios. Due to the Bayesian character of the analysis, the results still exhibited a distribution of probable parameter values showing the variability in the stream and between experiments. For each sampling date, we defined a threshold from the ambient conditions where the peak was considered to be completely retained. Measured values below that peak were removed for the analysis. By that, we removed cases, where accumulated measurement errors would exceed calculated retained amounts. Sampling date and leachate-source specific questions could be addressed by adding an experiment or leachate class variable as a random effect to the model.

Since the fluorescence of DOM increases linearly with concentration, we used $F_{max}$ of PARAFAC components analogously to concentrations in these models.

During a plateau addition experiment, concentration changes in a conservative tracer due to dilution effects can be described using Eq. (1). We used this equation to determine the dilution factors and to correct measured DOC and nutrient concentrations as well as DOM components by the measured changes in conductivity.

$$C_{x,t}=C_{amb,x,t}+\left(C_{x-1,t}-C_{amb,x,t}\right)\frac{dil_x}{dil_{x-1}} \tag{1}$$

x … index of longitudinal sampling points

t … index of addition date

$C_{x,t}$ … concentration at point x and date t(variable)

$C_{amb,x,t}$ … ambient concentration at point x and date t(variable)

$dil_x$ … dilution factor at point x (once calculated fixed values)

A reactive substance can be modelled using Eq. (2). Variable x from the original equation (Stream Solute Workshop, 1990) was replaced by $(d_{x-1}-d_x)$ to conform to a NARX problem.

$$C_{x,t}=C_{amb,x,t}+\left(C_{x-1,t}-C_{amb,x,t}\right)\frac{dil_x}{dil_{x-1}}e^{\frac{d_{x-1}-d_x}{s_w}} \tag{2}$$

$s_w$ … nutrient uptake length (parameter)

prior: $s_w \sim Lognormal(400,200), s_w \in [0.01,10000]$

$d_x$ … distance of point x from origin (fixed)



Using the flow velocity and the water depth, the nutrient uptake velocity can be calculated from $s_w$ (Eq. 3). This is useful to
reduce flow-dependent effects.

$$\frac{1}{s_w} = v_f (uz)^{-1}$$ (3)

$$C_{x,t} = C_{amb,x,t} + \left(C_{x-1,t} - C_{amb,x,t}\right) \frac{dil_x}{dil_{x-1}} e^{\left(d_{x-1} - d_x\right) v_f (uz)^{-1}}$$ (4)

$v_f$ … nutrient uptake velocity (parameter)

prior: $v_f \sim Lognormal(0.7,3), v_f \in [0.01,35]$

u … flow velocity (calculated by Hec-RAS, then fixed)

z … water depth (calculated by Hec-RAS, then fixed)

The areal uptake rate can then be modelled using Eqs. 5 and 6:

$$v_f = U C_{x,t}^{-1}$$ (5)

$$C_{x,t} = C_{amb,x,t} + \left(C_{x-1,t} - C_{amb,x,t}\right) \frac{dil_x}{dil_{x-1}} e^{\left(d_{x-1} - dx\right) U C_{x,t}^{-1} (uz)^{-1}}$$ (6)

U … areal uptake rate (parameter)

prior: $U \sim Lognormal(2,3), U \in [0.01,40]$

A linear relation between uptake velocity and concentration is needed to properly calculate U. In other cases, uptake
functions such as the Michaelis-Menten formulation can be used to describe the observed uptake-concentration relation
(Stream Solute Workshop, 1990). An uptake efficiency loss, mathematically described by a power function, was shown in
experiments with N-NO₃ (Dodds et al., 2002; O'Brien et al., 2007). A mechanistic argumentation for either of these
functions is difficult (Stream Solute Workshop, 1990), but testing the suitability with the Bayes factor leads to good
empirical fits.

To include interactions, we added a product of power functions for relevant compounds and nutrients (Eqs. 7 and 8). Where
beneficial, the wetted width w was added to incorporate influences of the stream bed surface on retention processes. For
positive exponents $m_i$ in equation (7), the function would pass through the origin. As this is not always true, we incorporated
an added value l as a degree of freedom. Biogeochemically interpreted, l > 0 means that the absence of a stimulating
component does not necessarily lead to a complete collapse of DOM or nutrient retention. The relevance of these effects was
tested in the modelling process by comparing different combinations of compounds in models using the Bayes factor.

$$v_f = kw \left(l + \prod_i C_{i,x,t}^{mi}\right)$$ (7)



$$C_{x,t} = C_{amb,t} + \left( C_{x-1,t} - C_{amb,t} \right) \frac{dil_x}{dil_{x-1}} e^{\left( d_{x-1} - dx \right) kw \left( l + \prod_i C_{i,x,t}^{mi} \right) (uz)^{-1}} \tag{8}$$

k … uptake rate factor (parameter)

prior: $k \sim Lognormal(0.7,3), k \in [0.01,35]$

w … wetted width, constant 1 to represent no influence (calculated by Hec-RAS, then fixed)

i … index of nutrient or DOM component

$C_{i,x,t}$ … concentration of compound i at point x and date t (variable)

$m_i$ … exponent determining the strength of the relations (parameter)

prior: $m_i \sim Normal(-0.2,0.4), m_i \in [-1,1]$ if a dampening influence was assumed from literature

$m_i \sim Normal(0.2,0.4), m_i \in [-1,1]$ if a stimulating influence was assumed

Since we had no prior information for $m_i$ from previous studies, it was important to test the influence of the prior on the final results by using a uniform distribution and normal distributions with different parameters. In the presented models, the priors

for any parameters did not dominate the results. The given limits for certain parameters were important for a stable model fit. Due to the double-exponential structure of Eq. (8) in $m_i$, the limits were essential for the convergence.

To set up the models, we used the difference of concentrations (Eq. 9) as the dependent variable and restructured the equations above accordingly. We assumed a normal error distribution for the differences of concentrations and the differences of fluorescence. The nature of the measurements would also allow a log-normal error distribution, but our data

clearly deviated from that assumption.

$$D_{x,t} = C_{x,t} - C_{amb,t} \tag{9}$$

$D_{x,t}$ … concentrations (DOC, SRP, N-NO₃) or fluorescence (DOM PARAFAC components) deviation from ambient conditions

model error assumptions: $D_{x,t} \sim Normal\left( \mu_{x,t}, \sigma^2 \right)$

$\mu_{x,t}$ … calculated difference from Eqs. (2), (4) and (8) restructured to suffice Eq. (9)


The accuracy of the model can be compared to expected measurement errors (e.g. lab instrument errors, errors from sampling procedure) and show the point where no additional information can be expected from the data (for proper error propagation analysis see Haefner, 2012, chapter 9). Using the simulated probability density of the residuals, which is in the same units as the measured values, we get an impression if further information can be expected from the data.

The 95% probability interval of the residuals can be a meaningful metric of the model accuracy. This approach makes it easier to distinguish between signal and noise compared to an approach where Eqs. (2), (4) and (8) are applied step-wise and error propagation is not considered. It can also help in planning the experimental scheme to improve the signal-to-noise ratio





because amongst others, the error depends on the instruments, sample handling, concentrations and concentration difference of consecutive samples.

We were interested if different leachate sources or dates would show different characteristics in $v_f$. A difference by leachate sources would show an influence from the source dependent quality difference. If the sampling date had an influence, we interpreted this as either a quality difference in different leachates from the same source or a not observed, date-related influence. This was done for each nutrient and DOM fraction by comparing the model using Eq. (4) to models using the same equation, but adding group-level effects for either the sources or the additions, of which there were two per source. The

comparison was done by means of the Bayes Factor. A Bayes factor larger than 1 means that a separate $v_f$ for each source or experiment date would increase the probability to observe the measured values. After finding the most suitable models using Eq. (8) we also compared these to the ones with group-level effects. This shows, whether the interaction term in Eq. (8) can cover or even outperform source or date related influences. By adding the group-level effects, a separate posterior distribution for each DOM source or each addition is produced and can be compared to each other.

For the comparison of uptake velocities between all nutrients and DOM fractions, we used a transformation of Eq. (4) to calculate $v_f$ for each nutrient and DOM component and between all pairs of sequent points directly. Uptake velocities between nutrients and DOM fractions were compared using a Bayesian test for linear correlation (Jeffreys, 1998; Ly et al., 2016) implemented in the R package BayesFactor (Morey et al., 2018).

## 3 Results

### 3.1 PARAFAC components

We could successfully fit a six-component PARAFAC model (Figure 3). We used Openfluor.org (Murphy et al., 2014) to compare and link the found components with other studies (Table 2). Leachates of pig and cow dung characteristically exhibited high levels of tryptophan-like (Trp, C5) and tyrosine-like (Tyr, C6) compounds. Leaf leachate showed high peaks in microbially produced humic-like (Hum-mic, C1) fluorescence, which is assumed to represent low-molecular, aliphatic

DOM originating from microbial degradation. Ambient water was characterized by humic-like material from terrestrial sources (Hum-ter, C2) and microbially processed terrestrial DOM associated with agriculture (Hum-micter, C3). Another humic-like fluorophore with some resemblance to pure quinone was identified in all sources (Qui, C4). The ambient DOM composition resembled the leachate from pig dung.

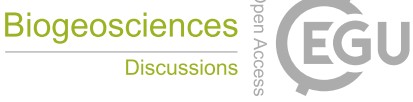

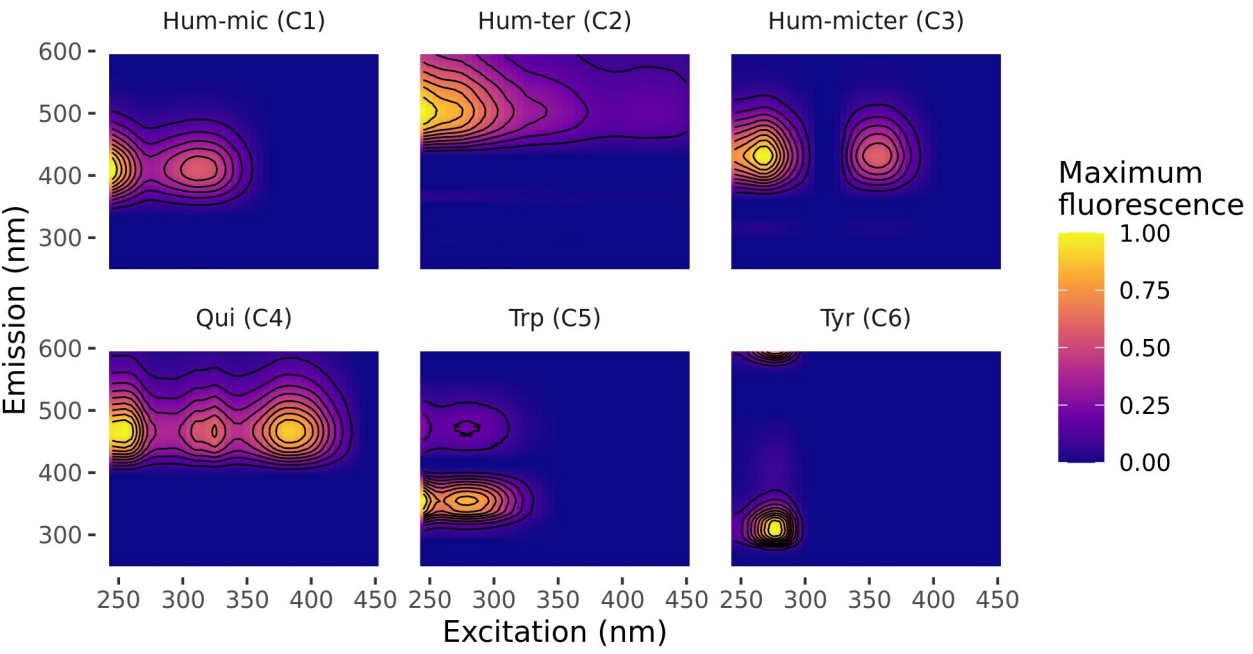

**Figure 3: Fluorescence spectra of the identified PARAFAC components.**

**Table 2: PARAFAC components and their comparison to other studies. The used abbreviations and symbols stand for: a: ambient, m: corn, c: cow dung, l: leaves, n: nettles, p: pig dung, ▲: high, ▬: intermediate, ▼: low.**

| component | similar components in other studies | interpretation | relative share in leachates | | | | | |
|---|---|---|---|---|---|---|---|---|
| | | | a | m | c | l | n | p |
| Hum-mic (C1) | G2 (Murphy et al., 2011), C2 (Lambert et al., 2016b), D2 (Shutova et al., 2014) | microbial humic-like, DOM produced during the microbial degradation of terrestrial DOM within freshwaters | ▬ | ▼ | ▼ | ▲ | ▼ | ▬ |
| Hum-ter (C2) | C2 (Lambert et al., 2016a), F3 (Heibati et al., 2017) | terrestrial humic-like, high molecular weight and aromatic compounds of terrestrial origin. | ▲ | ▼ | ▲ | ▼ | ▼ | ▲ |
| Hum-micter (C3) | C5 (Lambert et al., 2017), C4 (Williams et al., 2010), C5 (Williams et al., 2013) | microbial humic-like, positively correlated with bacterial activity and croplands in the catchment, associated with microbial transformation of terrestrial organic matter. | ▲ | ▼ | ▼ | ▼ | ▼ | ▲ |
| Qui (C4) | C2 (Yamashita et al., 2011), C2 (Garcia et al., 2015) | humic-like, A and C peaks, terrestrial origin, with an aromatic chemical | ▬ | ▲ | ▲ | ▬ | ▲ | ▬ |





| | | | | | | | | |
|---|---|---|---|---|---|---|---|---|
| | | nature, may be derived from old soil organic matter, some similarity to pure quinone. | | | | | | |
| Trp (C5) | C7 (Stedmon and Markager, 2005), C6 (Murphy et al., 2011) | tryptophan-like fluorescence, peak almost identical to free tryptophan, derived from autochthonous processes, correlated to terrestrial fluorescent material in forested catchments. | — | — | ▲ | — | — | ▲ |
| Tyr (C6) | G7 (Murphy et al., 2011), C3 (Yamashita et al., 2013), J3 (Wünsch et al., 2015) | tyrosine-like, is suggested as degradation products of peptides/proteins. | ▼ | — | ▲ | ▲ | ▲ | — |

## 3.2 Ambient concentrations and introduced material

Peak DOC concentrations were highest for cow dung leachate, followed by corn and leaves and lowest in nettles and pig
dung (Figure 4). Leachates of cow dung, pig dung and leaves showed the highest concentrations of SRP. The overall background concentrations of $N-NO_3$ were highly fluctuating, high in concentration, and hardly influenced by leachate addition. Most components declined during downstream travel, while Hum-ter (C2) and Hum-micter (C3) increased during corn and leaves additions. Concentrations and fluorescence tended to return to ambient conditions while travelling downstream. We calculated the correlation of DOC, $N-NO_3$, SRP concentrations and the fluorescence-based concentrations
of the DOM fractions (Table 3) to be aware of and avoid effects of collinearity on the models calculated in the further process.



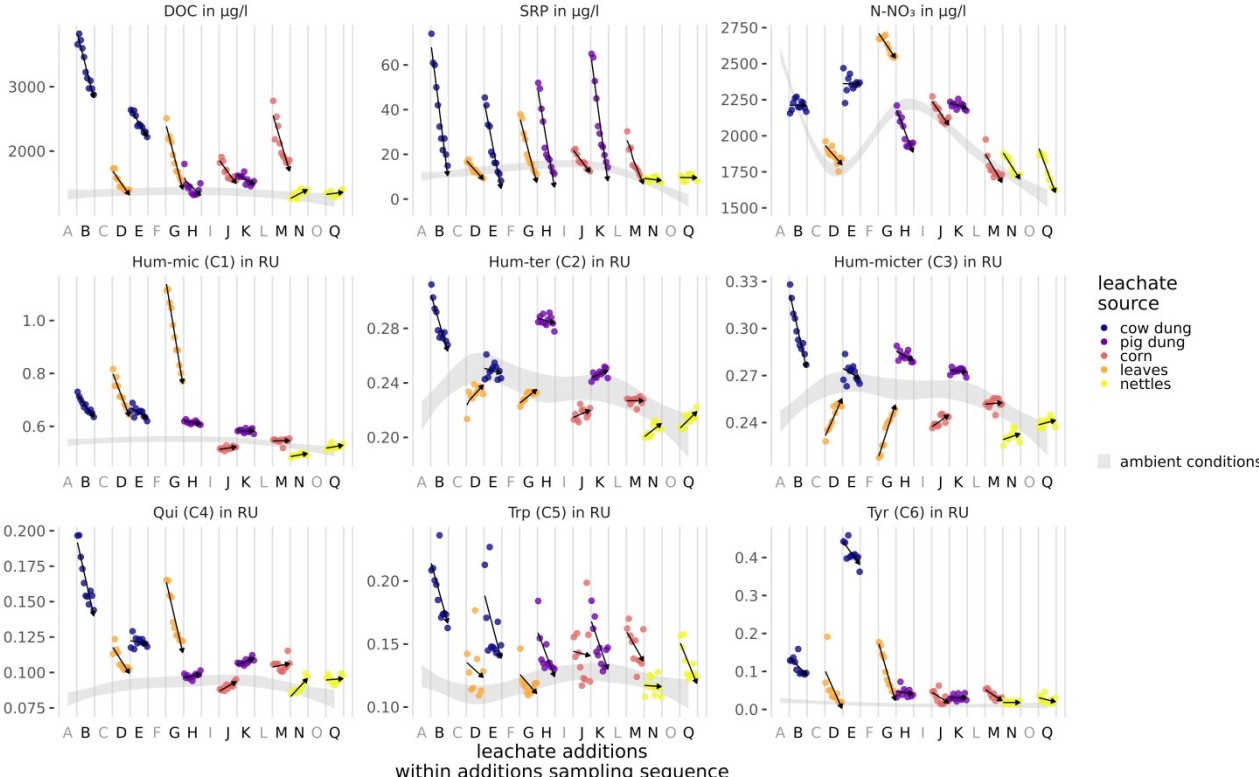

**Figure 4: DOC, SRP, N-NO₃ and DOM fractions as modelled in a PARAFAC analysis. The values are not corrected for dilution effects. Horizontally, the leachate addition experiments are shown as letter codes (see Table S1). Dates with no leachate addition are displayed as grey letters and the measured values are not shown. Each experiment (A to Q) is represented by a group of points and a trend arrow following the sequence of samples (earlier to later, up- to downstream). The ambient concentrations were interpolated from measurements taken in-between leachate additions and are visualized as grey ribbons (see Table S1 for ambient conditions and additional amounts from leachate additions at the upstream station). Vertically the concentrations of DOC, SRP and N-NO₃ and the maximum fluorescence in Raman units (RU) of the PARAFAC components are shown.**

**Table 3: Linear correlation of nutrient concentrations and DOM fraction fluorescence; Bayes factor in brackets; only shown, if Bayes factor > 1.**

|                     | Hum-mic (C1) | Hum-ter (C2) | Hum-micter (C3) | Qui (C4)     | Trp (C5)    | Tyr (C6)    | DOC         |
| ------------------- | ------------ | ------------ | --------------- | ------------ | ----------- | ----------- | ----------- |
| Hum-micter (C3)     | 0.87 (5.47)  | 0.62 (2.34)  |                 |              |             |             |             |
| Qui (C4)            |              | 0.86 (3.25)  | 0.59 (1.46)     |              |             |             |             |
| Trp (C5)            |              |              | 0.73 (2.45)     | 0.87 (8.22)  |             |             |             |
| Tyr (C6)            |              |              | 0.58 (1.03)     |              |             |             |             |
| DOC                 |              |              | 0.56 (1.38)     | 0.80 (12.62) | 0.91 (8.83) |             |             |
| SRP                 |              |              | 0.47 (1.18)     |              | 0.69 (4.74) | 0.37 (1.35) | 0.41 (1.99) |





### 3.3 Results from the INSBIRE approach

During the experiment, discharge varied (0.41 to 0.93 l s$^{-1}$) and we could clearly see more stable fitting behaviour using $v_f$

rather than $s_w$. As U changes with concentration (Dodds et al., 2002; O'Brien et al., 2007), we focused on $v_f$ during further

analysis and tested effects of different other parameters on $v_f$. By testing a linear relation, the Michaelis-Menten formulation

and a power function, we found the power function the most suitable one for the concentration-uptake velocity relations.

We calculated the distributions of DOC uptake velocities depending on the leachate sources (Figure 5). The probability

density of DOC from corn leachate, leave leachate and cow dung leachate was narrow, allowing for a clear distinction of $v_f$

between these three. Here, corn leachate was taken up fastest followed by leave and cow dung leachate. The probability

density of the uptake velocities of nettle and pig dung leachates was much broader than those of the other leachates, making

$v_f$ distinction more difficult. During nettles and pig dung leachate additions, the DOC peaks were lower and therefore

measurement errors have a higher influence. This demonstrates how a low number of observations or erroneous data

influences results in Bayesian statistics. Although we cannot make certain statements in relation to the other leachates, we

still see the probable range. In specific, we can assume that the probability of both uptake velocities exceeding 6 mm min$^{-1}$ is

very low and that pig dung leachate is probably taken up faster than cow dung leachate.

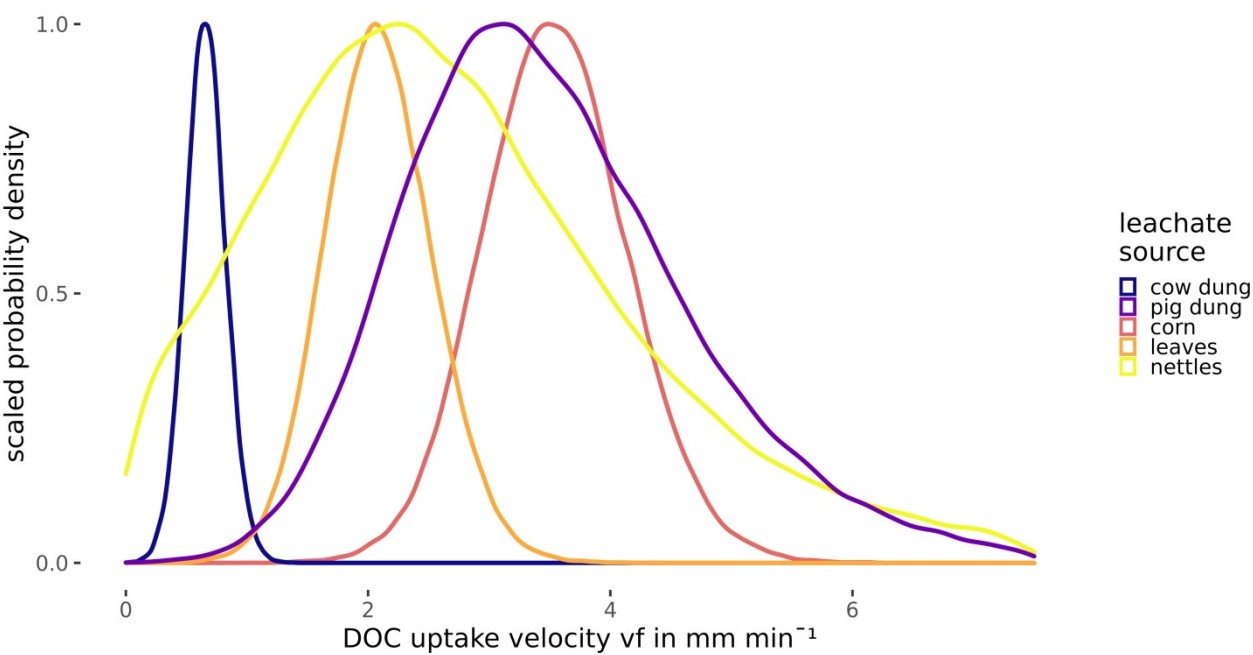

**Figure 5: Posterior density distribution curves of uptake velocity $v_f$ of DOC depending on the leachate source.** *Median $v_f$ in*
*mm min$^{-1}$ are:* **cow dung 0.66, pig dung 3.37, corn 3.54, leaves 2.08 and nettles 2.42.**





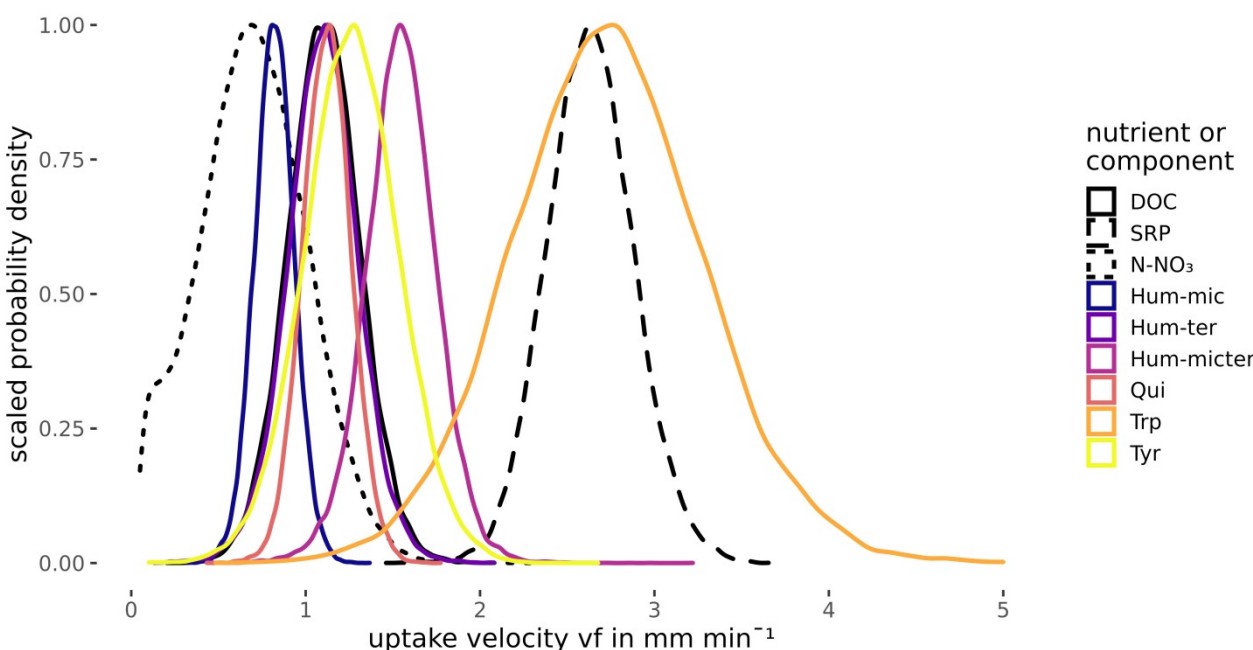

**Figure 6: Posterior density distribution curves of uptake velocity $v_f$ for different compounds and nutrients. Median $v_f$ in mm min$^{-1}$ are: DOC 1.11, SRP 2.63, N-NO$_3$ 0.73, Hum-mic (C1) 0.82, Hum-ter (C2) 1.10, Hum-micter (C3) 1.56, Qui (C4) 1.12, Trp (C5) 2.76, Tyr (C6) 1.27.**

Modelling $v_f$ of the different components and nutrients without any other considered influences showed that the uptake of the bulk DOC reflected the average uptake of the different DOM components. N-NO$_3$ and Hum-mic (C1) were taken up slower, SRP and Trp (C5) were taken up faster than the bulk DOC (Figure 6).

Differences between samplings using the same source can also be caused by other day-dependent characteristics such as discharge or weather. Hence, we tested whether the addition date of the different experiments significantly affected the
uptake of the DOM components or nutrients. Hum-mic (C1) retention was substantially (BF > 3.2) and Trp (C5) and Tyr (C6) retentions were decisively (BF > 100) influenced by the addition date. Bulk DOC retention was decisively influenced by the DOM source. The source also strongly influenced Tyr (C6) retention and the addition date had a decisive influence on the DOC retention, but both were outperformed by the respective other effect. Hum-ter (C2), Hum-micter (C3) and Qui (C4) showed conservative uptake behaviour independent of the source or addition date (BF < 1, Table 5).


**Table 4: Model comparison $v_f$ with and without random effects (mixed models, MM) of source and additiondate. The Bayes R$^2$ was calculated to show the absolute model performance and the Bayes Factor was used to tell whether adding information leads to a model improvement.**

|  | vf | MM source | | MM addition | | comment |
|---|---|---|---|---|---|---|
| model | Bayes | Bayes | BF vf | Bayes | BF vf | |





| | R² | R² | vs. | R² | vs. | |
|---|---|---|---|---|---|---|
| Hum-mic (C1) | 0.51 | 0.48 | 0.17 | 0.50 | 4.61 | The addition date has a substantial impact on Hum-mic (C1) degradation. |
| Hum-ter (C2) | 0.34 | 0.49 | 0.7 | 0.49 | 0.65 | Neither addition date nor source improved the model. |
| Hum-micter (C3) | 0.52 | 0.51 | 0.21 | 0.54 | 0.22 | Neither addition date nor source improved the model. |
| Qui (C4) | 0.46 | 0.46 | 0.09 | 0.45 | 0.12 | Neither addition date nor source improved the model. |
| Trp (C5) | 0.29 | 0.29 | 1.39 | 0.48 | 134.23 | The addition date has a decisive influence on the Trp (C5) degradation. |
| Tyr (C6) | 0.29 | 0.34 | 10.68 | 0.70 | 1.2e8 | Both, source and addition date improved the model. The effect of sampling was decisive. |
| DOC | 0.26 | 0.46 | 1563 | 0.46 | 146 | The DOM source has a decisive influence on the DOC degradation. While the addition also has a decisive influence, it is rejected due to a higher complexity and lower probability of the model including the source. |
| NO₃ | 0.16 | 0.29 | 0.41 | 0.29 | 0.65 | Neither addition nor source improved the model. |
| SRP | 0.56 | 0.57 | 0.17 | 0.56 | 0.11 | Neither addition nor source improved the model. |

To disentangle the interaction effects between nutrient and DOM component uptake velocities, we used Eq. (8) to fit the parameters to the measured data. Since Eq. (8) describes the absolute nutrient retention, we inserted the posterior probabilities of the parameters into Eq. (7) to analyse and interpret changes and interactions in the uptake velocity and produce Figure 7. The fitted parameters ($k$, $m_i$) as well as the measured concentration ranges were necessary to reveal the strengths, shapes and probability intervals of the interactions. We found the following interactions (Table 5, Figure 7).

DOC uptake velocity was lower at higher concentrations of Tyr (C6), but there is strong evidence that the leachate source variable offers a better explanation. The SRP uptake velocity increased with higher wetted width and was lower during high SRP concentrations. The uptake velocity of Hum-mic (C1) was higher with a broader wetted width and at lower concentration (Table 5). Including these terms improved the Hum-mic (C1) model even more than including the addition date (Table 3). Adding group level effects for the addition date to $k$ of Eq. (8) did not improve the model further. Therefore,

the addition date acted as a surrogate variable for the wetted width and the Hum-mic (C1) concentration, but could not explain the retention equally well. Hum-ter (C2) retention was stimulated by the DOC concentration. The Qui (C4) retention was dampened by itself and Hum-mic (C1). The Trp (C5) retention was dampened by itself (efficiency loss) and Hum-ter (C2), but could not outperform the model with the sampling date included. Tyr (C6) was retained slower with higher fluorescence in itself and Hum-ter (C2). Although the model improved decisively in comparison to the one without

interactions, it could not exceed the model with the sampling date in probability. Other than in the Hum-mic (C1) model, the sampling date variable still contained more important information than the interactions found for Trp (C5) and Tyr (C6)





uptake velocity. For Hum-micter (C3) and $NO_3$, no additional information could be gained from the available data. We found

no effects of variable collinearity within the models (Table 5, Table 3, Table S2).

We analysed correlations between uptake velocities of nutrients and different DOM components to check for concurrent

retention, which might indicate interrelations among or dependencies of different microbial metabolic processes, such as,

e.g., the combined need of these substances in the microbial metabolism (Table S2). We found substantial evidence that $v_f$ of

Qui (C4) correlated with $v_f$ of Tyr (C6) and DOC, indicating that the retention of Qui (C4) concurred with Tyr (C6) and

DOC.

**Table 5: Interactions between uptake velocity and concentrations of other nutrients or DOM components using the INSBIRE**
**approach. $v_f$: uptake velocity, k uptake rate factor, w: wetted width, $C_i$: fluorescence of PARAFAC components, $m_i$: exponent of**
**relation, l: additive parameter**

| fraction/ nutrient | most probable model (Eq. 7) | Bayes $R^2$ | BF vs. vf | test variables | estimates, [95% probability interval] |
|---|---|---|---|---|---|
| Hum-mic (C1) | $v_f = k\, w\, C1^{mc1}$ | 0.60 | 16.74 vs. addition: 3.6 | P(w ≠ 1): BF = 7.34 P(mc1 ≠ 0): BF = 1.4 | k = 2.11, [1.65, 2.59] mc1 = −0.38, [−0.93, 0.28] |
| Hum-ter (C2) | $v_f = k\,(l + DOC^{mc})$ | 0.34 | 7.69 | P(l ≠ 0): BF = 2.36 P(mc ≠ 0): BF = 7.69 | k = 0.11, [0.01, 0.61] l = 3.16, [0.23, 8.01] mc = 0.32, [−0.42, 0.60] |
| Hum-micter (C3) | $v_f = v_f$ | - | - | - | - |
| Qui (C4) | $v_f = k\, C1^{mc1}\, C4^{mc4}$ | 0.44 | 3.13 | P(mc1 ≠ 0): BF = 2.54 P(mc4 ≠ 0): BF = 2.44 | k = 0.71, [0.14, 2.23] mc1 = −0.25, [−0.89, 0.39] mc4 = −0.35, [−1.05, 0.38] |
| Trp (C5) | $v_f = k\, C2^{mc2}\, C5^{mc5}$ | 0.30 | 3.87 vs. addition: 0.03 | P(mc2 ≠ 0): BF = 2.71 P(mc5 ≠ 0): BF = 3.13 | k = 0.85, [0.10, 3.20] mc2 = −0.44, [−1.23, 0.35] mc5 = −0.55, [−1.31, 0.22] |
| Tyr (C6) | $v_f = k\, C2^{mc2}\, C6^{mc6}$ | 0.45 | 1.51e7 vs. addition: 0.12 | P(mc2 ≠ 0): BF = 2.34 P(mc6 ≠ 0): BF = 1.46e7 | k = 0.27, [0.06, 0.76] mc2 = −0.23, [−0.98, 0.52] mc6 = −0.96, [−1.25, −0.69] |
| DOC | $v_f = k\, C6^{mc6}$ | 0.28 | 10.50 vs. source: 0.01 | P(mc6 ≠ 0): BF = 10.50 | k = 0.30, [0.10, 0.75] mc6 = −0.62, [−0.95, −0.18] |
| $NO_3$ | $v_f = v_f$ | - | - | - | - |
| SRP | $v_f = k\, w\, SRP^{mp}$ | 0.63 | 1.45e4 | P(w ≠ 1): BF = 31.93 P(mp ≠ 0): BF = 6.21 | k = 26.18, [10.17, 39.20] mp = −0.31, [−0.45, −0.07] |





**Figure 7: Simulated change of uptake velocity $v_f$ with variation of one parameter Table 1. The colours show the 50 % (violet) and the 90 % (yellow) percentile intervals.**





The simulated probability density of the residuals (figure S1) was compared to the expected accuracy of the instruments for DOC and SRP. For a straightforward impression, we neglected errors in the exponent. Following this, the models depended on three measured values ($C_{x,t}$, $C_{amb,t}$, $C_{x-1,t}$). Thus, we multiplied the instrument errors by 3 to get the accuracy of the model based on the instrument accuracy. For DOC samples around 2000 µg l$^{-1}$, this would be 120 µg l$^{-1}$. The 95% probability interval of residuals of the DOC model (mixed model including leachate source) was between −172 and 131 µg l$^{-1}$. Given

additional unknown errors from the sampling procedure, there was little more information to be expected from the data. In contrast, the instrument accuracy for SRP multiplied by 3 was 0.3 µg l$^{-1}$, and the 95% probability interval of the residuals was between −4.74 and 4.85 µg l$^{-1}$ for the model with wetted width and SRP concentration included in the exponent (Table 5). This shows that the model for SRP has still potential for improvement by, e.g., adding meaningful variables not measured in this study or by increasing the number of observations. A similar analysis of the PARAFAC components is not as simple

because there is no conventional way of calculating the accuracy of a PARAFAC model's sample loadings.

## 4 Discussion

### 4.1 Uptake of bulk DOC

The uptake velocity of bulk DOC varied between leachate sources (Figure 5), as also observed in previous studies (e.g. Bernhardt and McDowell, 2008; Mineau et al., 2016; Mutschlecner et al., 2018). Experiments with leachates from different

natural organic matter in streams are scarce, and a clear picture cannot be drawn from the published literature. Concerning anthropogenic and natural sources, we could observe a slower uptake velocity for the DOC from cow dung leachate in comparison to leaves and corn leachates. Although corn is not occurring naturally in this area, the derived DOM is comparable to the leachate of local tree leaves indicated by the similarity in DOM components (Table 2). To our knowledge, there was only one leachate addition study working with manure (originating from cow, Kuserk et al., 1984; uptake velocity

calculated in Mineau et al., 2016). They observed a median uptake velocity of 0.31 mm min$^{-1}$, while we observed a median of 0.66 mm min$^{-1}$. Our results were within the observed range of reported uptake velocities. Due to a broad and overlapping posterior distribution, we could not make any inference about the nettles leachate. Also, the pig dung leachate showed a broad posterior due to little data but was definitely degraded faster than the cow dung leachate. We could see a similarity between the ambient DOM quality and the pig dung leachate. This might stem from the pig dung allied as fertilizer in the

catchment. We suggest a potential adaption of the microbial community to this DOM quality, which results in a high $v_f$. The median $v_f$ of the leaf leachate was 2.08 mm min$^{-1}$ and slightly higher than the median of 1.29 mm min$^{-1}$ identified within eight studies (Bernhardt and McDowell, 2008; Kaplan et al., 2008; McDowell, 1985; McDowell and Fisher, 1976; Meyer et al., 1988; Mineau et al., 2013; Mutschlecner et al., 2018, Hall and Baker unpublished) and summarized by Mineau et al. (2016) and the $v_f$ of 1.22 mm min$^{-1}$ reported by Graeber et al. (2019) for Alder leaf leachate in an agricultural stream. To our

knowledge, there was no uptake velocity for corn or nettle leachate explicitly published so far.



We found a relation of the bulk DOC uptake velocity to the Tyr (C6) fluorescence (Figure 6 a) when calculating a source-independent model. Still, the mixed effects model with the leachate source included performed much better. This indicated that, apart from the fluorescence of Tyr (C6), other, probably non-fluorescent, components influenced the bulk DOC uptake, which we could not detect with our methods. We expected no influence on the DOC retention by N-NO₃, which was not a

limiting nutrient due to its high concentrations. However, we could not find evidence for an influence of SRP concentration either, although there is evidence that DOC uptake is stimulated by P in P-limited systems (Mutschlecner et al., 2018). The SRP concentrations were not intentionally raised in our study and showed a P limitation according to the Redfield ratio in 92% of the measurements. Besides, DOP in the leachates could have acted as another P source but was not measured. Thus, SRP-related effects in DOC retention might have stayed uncovered.

**4.2 Uptake of DOM and nutrients**

The various DOM fluorophores were retained with different uptake velocities, but the uptake velocity density curves exhibited more or less broad ranges with overlaps (Figure 4). Therefore, we did not find a strict fluorophore-based bioavailability in our experiment. In general, the bioavailability of a fraction is not only depending on the chemical composition, but also on the ecosystem and the involved microbial community (Kamjunke et al., 2015), the overall

availability of different fractions and nutrients (Berggren and Giorgio, 2015; Bernhardt and McDowell, 2008; Mutschlecner et al., 2018) and transport characteristics (Ejarque et al., 2017). We performed the experiments in a small homogeneous stretch of a stream and already found considerable variability in DOM fluorophore-specific uptake. Therefore, we would expect even more variation in hydromorphologically different stretches, streams or different seasons.

In our study, Hum-mic (C1) was taken up slowest, while Trp (C5) was taken up fastest, similar to SRP. The fast uptake, we

observed for Trp (C5), was also found in previous studies for different amino acid-like fractions (Findlay and Sinsabaugh, 2003). In contrast, the uptake velocity of Tyr (C6) was not specifically high. This might be caused by a release of Tyr (C6) as a degradation product of humic substances (Stevenson and He, 1990; Tsutsuki and Kuwatsuka, 1979). The fast uptake of SRP supports our impression of P being a limiting factor although some P was introduced by the leachate additions.

In all DOM fractions but Hum-micter (C3), we found at least a substantial dependence of the uptake on other variables and

self-dampening effects of uptake. Lower uptake velocity with increasing concentration, interpreted as efficiency loss, was previously described for nitrogen (Dodds et al., 2002; O'Brien et al., 2007). A similar self-dampening effect could be shown for Hum-ter (C1), Qui (C4), Trp (C5), Tyr (C6) and SRP as well. These effects can be explained by a specific processing capacity of the stream ecosystem. This capacity is influenced by adaption to usually occurring concentrations (Fasching et al., 2020; Tihomirova et al., 2012) and transport limitations (Weigelhofer et al., 2018a, 2018b). Hum-mic (C2), Hum-micter

(C3) and DOC retention showed no evidence of efficiency loss (BF was around 1) at the measured concentrations, indicating the stream was able to retain more without a decline in uptake velocity. So far, we have not found any other studies presenting efficiency loss for DOM fractions.





Additionally to self-dampening, we also observed dampening effects among different components. Interactions in uptake processes can have different reasons and are, therefore, difficult to interpret. Stimulating interactions can arise, e.g., from the

stimulation of the uptake of one substance by the presence of another through priming (but see critical discussion in Bengtsson et al., 2018). Dampening interactions can be caused by the preferential uptake of one fraction over another (Brailsford et al., 2019) or inhibitory effects between different substances (Freeman et al., 1990). Furthermore, the degradation of DOM can cause one molecule to break down into other ones and can cause an increase of a fraction, while another one decreases (Kamjunke et al., 2017). In our study, we mainly observed dampening effects among different

components. As preferential uptake should have caused negatively correlated uptake velocities of the involved fractions, which were not found (Table S2), we assume that the observed dampening effects were mainly caused by decomposition from one DOM component into another. Substances with a low degree of humification contain a significant amount of amino acids, including tyrosine and tryptophan, as well as quinones (Kamjunke et al., 2017; Stevenson and He, 1990; Tsutsuki and Kuwatsuka, 1979), which can be separated during degradation.

In our study, Qui (C4) was degraded slower at higher Hum-mic (C1) fluorescence. The molecular structures found in the literature (Stevenson and He, 1990; Tsutsuki and Kuwatsuka, 1979) suggested that Qui (C4) is a product of the Hum-mic (C1) degradation and its net retention was, therefore, dampened by a concurrent production. Similarly, Trp (C5) and Tyr (C6) might have been degradation products of Hum-ter (C2). Hum-ter (C2) seemed to need energy in the form of carbohydrates or other essential components for the degradation because the DOC concentration stimulated its uptake. We

saw a weak probability, that the uptake velocity of Hum-ter (C2) was stimulated by Qui (C4, BF = 1.9) and Tyr (C6, BF = 1.8). Due to the broad shape of the fluorescence spectrum, we conclude, that Hum-ter (C2) is a heterogeneous fraction. Therefore, several combined processes and effects might have been responsible for the observed uptake patterns. Only a part of the degradation seemed to be stimulated by other DOM fractions, which we concluded from the importance of an additive value l in the model (Table 5). This result also supports the hypothesis of a heterogeneous fraction.

We found substantial evidence that Qui (C4) was degraded simultaneously with Tyr (C6) and bulk DOC. General degradation conditions, such as low transport limitation (Weigelhofer et al., 2018b) or stretch-wise more productive microbial communities, can foster simultaneous turnover (Guillemette and Giorgio, 2012). Also, favourable stoichiometric ratios for microbial metabolism can stimulate concurrent degradation. We consider concurrent degradation and interactions essential characteristics of the complex DOM degradation processes. With the data at hand, we cannot favour any of these

hypotheses, but INSBIRE indicated that there is a concurrent behaviour of Qui, Tyr and bulk DOC, and further experiments may help to elucidate, which of the proposed mechanisms is responsible.

Our model also revealed some hydromorphological effects on DOM fluorophore and nutrient uptake. The wetted width could partly explain the uptake of Hum-mic (C1) and SRP. We interpret this as an influence from sorption to sediments or uptake by the benthic microbial community. The adsorption of humic substances to clay is generally strong when the ionic

strength is high (Theng, 2012). The conductivity around 630 µS cm⁻¹, which was measured during the experiment, as well as the clay-dominated sediments offered good conditions for adsorption. Therefore, we inferred that Hum-mic (C1) and SRP





were partly adsorbed to clay particles in the stream sediment, and we can see this in the importance of the wetted width on their uptake velocity. For Hum-micter (C3), there was weak evidence (BF = 1.7) that the wetted width explains the retention as well, but for all other nutrients and DOM fractions, an influence was unlikely (BF < 1). The component-specific influence

of wetted width suggests a DOM quality dependent localization of uptake processes in our study. Contrary to the common assumption that uptake processes are dominated by the benthic community (Battin et al., 2016; Wiegner et al., 2005), Graeber et al. (2018) and Kamjunke et al. (2015) proposed a potentially important impact of planktonic bacteria on in-stream DOM uptake processes. In our study stream, such planktonic uptake might be dominating for the uptake of all DOM fractions except Hum-ter (C1), where the substantial influence of wetted width indicated an importance of the benthic

community.

## 4.3 Potential and limitations of the INSBIRE approach

The INSBIRE approach was developed after the data from the experiment was acquired due to limitations in other data analysis methods developed for inorganic nutrient uptake (Stream Solute Workshop, 1990), such as the lack of a strategy to handle interactions among DOM components. Thus, our study represents a case study for the application of INSBIRE in the

analysis of DOM uptake, but does not claim to be a systematic check of the developed approach. Using INSBIRE for our experimental data helped to reveal novel interactions in DOM and nutrient uptake characteristics and also provided some information about the potential, but also the limitations of this method. Nevertheless, an application under controlled laboratory conditions is still open to thoroughly test the INSBIRE approach.

The underlying concepts, such as nutrient spiralling (Stream Solute Workshop, 1990), NARX models (Billings, 2013;

Leontaritis and Billings, 1985) and Bayesian statistics, have been investigated and developed for at least some decades. With this available knowledge, it was possible to develop the approach on a solid theoretical basis and with already existing concepts and algorithms. INSBIRE can be adapted by changing the underlying equations, using different solving schemes, and using different kinds of data. We used fluorescence measurements to determine the DOM quality, but INSBIRE is capable of incorporating any other data of different solvents (e.g. toxins or pesticides) and methods (e.g. mass spectroscopy,

liquid chromatography). The power function has proven useful in our study, but the approach facilitates the use of other equations if suited better for the respective case. Due to the formal description of the uptake processes, extrapolations to different ambient or event-related concentrations can be done (Payn et al., 2005).

The presented plots of the $v_f$ posterior density curves are intuitive to interpret and can help in our understanding and perception of the retention processes. The presentation in form of probability distributions rather than single values

corresponds to the experience that ecosystems are inhomogeneous while still assessable (McCarthy, 2007). For further studies, these posterior density curves can be directly used as prior information for similar models.

During the analysis, we found evidences, although weak, for even more interactions than presented here. The Bayesian nature of the analysis allowed us to evaluate even such weak relations, and we think it would be worth to test these in further



experiments. Also, we could show the limitation of the DOC retention model due to the accuracy of the measurements and
the heterogeneity of the measured molecules.

When a small number of observations is available, but the general knowledge about a topic is profound, it is possible to include data from previous studies as well as expert knowledge by means of non-conservative prior densities of the parameters. Then, results can be more precise and decisions can be based on both measured data and other available knowledge (Kuhnert et al., 2010; Lemoine, 2019). Even a low number of observations may show certain trends in DOM
uptake (Figure 5), which might be especially useful for monitoring or management decisions.

## 5 Conclusion

Human impacts, such as agricultural land use or wastewater discharges, have changed the quantity and composition of terrestrially derived DOM in streams ecosystems. Our study demonstrates that in-stream DOM uptake is source-depended and, thus, influenced by DOM quality, although we did not observe any significant correlations between bulk DOC uptake
and those of DOM components, such as co-leached nutrients or specific fluorophores. One reason for this lack of correlation could be that DOM uptake comprises a variety of simultaneously or sequentially occurring microbial degradation and production processes. The presented INSBIRE approach provided evidence for interactions among different DOM components, which indicate transformations of one substance into another during DOM processing. Besides, identification of different DOM components via spectroscopic characterization may be too imprecise to reveal the influence of DOM
components on DOM uptake, either because different molecules show similar fluorescent peaks or because of non-fluorescent components influence bulk uptake. Thus, further studies on DOM processing under controlled conditions are required which identify important molecular groups, such as, e.g., amino acids, sugars, or humic acids, more accurately.

Our study also shows that the uptake of bulk DOC, but also that of specific DOM components may be subject to efficiency loss, so far only known from nutrient uptake. This means that the uptake efficiency declines with increasing concentration of
the respective component. However, individual DOM components were not equally affected by efficiency loss or interactions with other components, indicating that the component-specific uptake capacity of benthic biofilms may depend on the respective microbial processes involved. Further studies need to look more closely into the underlying mechanisms of both efficiency loss and component interactions during DOM processing in aquatic ecosystems. Here, the developed INSBIRE approach may help to find concurrent retention and interactions of DOM components, thus providing an efficient
tool for the analysis and the management of organic carbon cycling in aquatic systems affected by human impacts.

## Code availablility

The codes necessary for applying the INSBIRE approach can be downloaded from https://doi.org/10.5281/zenodo.4071851 (Pucher, 2020).

**Author contribution**

Conceptualization: MP and GW; formal analysis: MP; funding acquisition: GW; investigation: MP, PF; methodology, MP, GW, KF, PF; project administration: GW; software: MP; supervision: TH, KF; validation: DG; writing – original draft and preparation: MP, GW; writing – review and editing: all co-authors.

**Competing interests**

The authors declare that they have no conflict of interest.

**Acknowledgements**

We would like to thank Verena Winiwarter and Astrid Harjung for their inputs on the text and the HR21 faculty and students for their thoughts in several discussions. The field and lab work would not have been possible without the help of Lenardo Zoltan, Yu-Ting Shi, Nikloaus Schobesberger, Isabella Fischer, Elmira Akbari, Ewelina Sonnenberg, André Fonseca, Annette Puritscher, Gertraud Stenicka and Beate Pitzl. Additionally, we would like to thank the staff from the Bundesamt für
Wasserwirtschaft in Petzenkirchen, Austria, especially Gerhard Rab, Alexander Eder and Günther Schmid for their help at the HOAL stream.

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
