# Peer review of "Complex interactions of in-stream DOM and nutrient spiralling unravelled by Bayesian regression analysis"

_Biogeosciences, 2020_

## Referee Comment (RC1) · Anonymous Referee #1 · 27 Nov 2020

General Comments:

The manuscript titled "Complex interactions of in-stream DOM and nutrient spiralling unravelled by Bayesian regression analysis" compares instream uptake of DOM differing in source/quality which has been relatively understudied in the literature. The authors pose an interesting research question well within the scope of biogeochemistry. A secondary objective of the manuscript was to refine a statistical model to estimate nutrient uptake that can provide estimates of uncertainty, account for nonlinearity, and allow for the addition of different nutrient fractionations (DOM optical properties here) to examine differences in uptake. The INSBIRE model appears robust and useful, but

its mathematical/statistical evaluation is outside of my expertise. I appreciate the value in the author's research, but my opinion is that this manuscript needs major revisions before publication here or elsewhere.

1. My major concern is the lack of independence in the study design. Nutrient/DOC leachate additions were added to the same stream reach several times per week. Subsidies from these additions are going to stimulate periphyton communities increasing their metabolic activity and biomass. More metabolic activity/biomass of bacteria and algae is going to result in faster uptake velocities of DOC and nutrients (e.g., SRP). Thus, I do not think it is entirely fair to compare uptake rates of different leachates between multiple additions that occurred over a few weeks as periphyton communities would have had ample time to use these resource subsidies to increase production and potentially alter subsequent nutrient uptake measurements. Commentary is needed in the methods to justify the study design and in the discussion to address the potential effects of repeated resource subsidies on nutrient uptake rates.

2. While I see the value in the INSBIRE model, I think there is too much commentary on it in the manuscript for this type of journal. I think these sections could be simplified to improve flow and keep the focus of the manuscript on DOM dynamics – as was outlined by the research questions in the introduction. Much of the detailed information could be added to a supplement for interested readers and anyone wanting to use the ISBIRE model in their own research.

a. The used of probability distributions to describe uncertainty is a valuable aspect of Bayesian modelling and INSBIRE. However – while visually appealing – Figures 5 and 6 are hard for the reader to interpret (e.g., how much do they overlap). The use of numerical 95% or 90% credible intervals instead of these distributions would be beneficial to the reader as the degree of overlap can be readily ascertained.

3. This manuscript would greatly benefit from a thorough editorial review to improve sentence structure, clarity, and flow. Some of the more complex sections in the methods, results, and discussion were hard to follow making it difficult to understand what was done and provide a comprehensive academic review of the manuscript . I have included some technical corrections below, but level of editing needed is beyond my capacity as a reviewer and my editorial comments are not complete.

a. Paragraphs in the discussion and introduction would benefit from clear introduction and conclusion sentences to define the topic in the body. Clarity in structure would help the reader follow along.

b. Avoid starting sentences with "it", "these", "this", "they" ect. especially when multiple subjects are being referred to. Best to be specific and clear to help the reader follow along.

Specific/Technical Comments:

Introduction

1. The content of the introduction is good, but a thorough editorial review is needed to help the reader follow along and increase the connection between the content/concepts introduced.

a. Could add a bit more information on DOM composition e.g., mixture of labile and recalcitrant compounds to the paragraph on lines 56-69.

2. A better definition of dampening/stimulating effects is needed, and I do not think this is the most appropriate term. Nutrient uptake by stream communities has an upper limit simply due to scaled up enzyme kinetics. Once that uptake rate is reached, it will not increase any further even with increased nutrient concentrations. Your non-linear models are entirely appropriate, but the way that "dampening effects" are defined (line 80: "dampening effects of nutrient concentration on the uptake efficiency") and referred to throughout the manuscript is not clear to the reader. The lack of clarity here continues in the discussion in the paragraph on "dampening effects".

Line 42: sentence is vague and could likely be connected to the following sentence to

add context.

Line 43: relate – is related to; stream and rivers are synonyms maybe differentiate with more context e.g., headwater streams, large rivers

Line 45-46: duplicate references

Line 46-47: DOM influences the toxicity of pesticides has no context and does not relate to your research question, consider removing – also duplicate references

Line 48: "this capacity" – what capacity? Can link to the previous sentence to add context

Line 61: is produced – are produced

Methods

1. Section 2.1 additional commentary – a sentence or two – on the Hydrological Open Air Laboratory would be better than just a citation.

2. Section 2.2 see general comment on experimental design. Increased justification is needed.

3. I think sections 2.6 and 2.7 need to be simplified and organized based on how you presented the research questions in the introduction to make the methods easier to follow. Two models were used to look at 1) random effects of DOM quality and 2) interactions? Clear definition of variables in the interaction models would be helpful.

a. For interactions did you only used the power function with your independent variables? Did you test different functions or just use the power one? I guess the choice of the power function was because it was used in previous studies. I think an exponential function or a logistic function, if you want to go one step further, is more ecologically appropriate.

b. I am confused by the addition of wetted width as an interaction. Sure, more surface

area might equal more retention of DOM, but all measurements occurred in the same stream with little difference in discharge.

Throughout: units for liters should be L

Results

1. The result section had many method-like statements in it. Some of these statements provided information that was not present in the methods. A clear methods section that follows the research questions presented in the introduction would help to keep the results section as a summary of what was found.

2. Results could be better structured around the 3 research questions from the introduction.

Discussion

1. Discussion could be better structured around the 3 research questions from the introduction

2. Stronger introduction sentences are needed set up what is going to be discussed in each paragraph.

3. "Interactions among different DOM components, which indicate transformations of one substance into another during DOM processing" – an interaction means that the relationship between uptake and concentration of one component is dependent on the concentration of another component. For example, more SRP leads to a more rapid increase in the uptake of a component relative to concentration. The interaction models were difficult to understand, but the above conclusion does not make sense. If DOM was being broken down into different components, the uptake of one component would be positively associated with the concentration of a degradation product. Since all leachate additions differed in composition the concentration of individual components is confounded. Correlations among net uptake of the different components may be better suited to address this.

Line 403-405: these two sentences contradict each other. Provide more context.

Line 403-420: This paragraph goes back and forth comparing uptake within your study and between your study and others. I suggest sticking with the latter and write a new paragraph for the between leachate differences you observed in your study. This new paragraph will also help transition into the next paragraph.

Line 414: dung allied – manure added

Line 421: relation – relationship

Line 432: should this be Figure 6?

Section 4.2: first paragraph states that there was no major difference in DOM bioavailability indicated by the broad overlap of parameter ranges, but the second paragraph discusses differences in uptake of different DOM components. The two paragraphs contradict each other.

---

## Referee Comment (RC2) · Anonymous Referee #2 · 1 Dec 2020

In their manuscript entitled "Complex interactions of in-stream DOM and nutrient spiralling unravelled by Bayesian regression analysis", Pucher et al. investigate the interactions between DOM and nutrient uptakes in a small stream based on an experimental setup. They used five different leachates having contrasting DOM properties based on optical measurements (PARAFAC), added these leachates into the stream and then measured DOC and nutrient concentrations and DOM properties along a 215 m reach. For interpreting their data and calculate uptake velocities, they proposed a new approach based on the spiralling concept and called Interactions in Nutrient Spirals using BayesIan REgression (INSBIRE).

[Figure]

The topic is of great interest, however the manuscript is hard to follow for readers that are not familiar with this type of approach. Furthermore, some clarification are required regarding the relevance of INSBIRE. Indeed, it seems to me that the model requires a lot of parametrization that is subjective (e.g. lines 365-378 or lines 390-400). It is like turning buttons to fit each data individually without clear ideas about the processes behind. It is therefore hard to interpret the data, as recognized by the authors (line 454 for instance) that finally can only make hypothesis on the processes occurring (e.g. lines 482-487). It doesn't seem to me that INSBIRE allows finally to investigate or quantify properly the interactions between DOM and nutrients uptakes, and I was also wondering to which extent this approach could be used by other researcher and/or in other study sites.

Introduction :

The introduction lacks of context regarding the interactions between DOM and nutrients, and why this is an important issue. The two first paragraphs are very broad, focusing on the importance of DOM on the biogeochemical and ecological functioning of freshwater ecosystems and on the impact of agriculture on DOM (which is not the subject of the study), and do not help the reader to understand why "the effects of changed DOM and nutrient supply on the DOM and nutrient uptake in streams remains in the dark" (lines 54-55) or why the authors "expect a complex interaction between the different DOM fractions and the available N and P to explicate the bioavailability and the aquatic retention of the DOM" (lines 66-67). The introduction could be improve by including more context about DOM/nutrient interactions based on previous works (e.g. Guillemette and del Giorgio 2012; Vonk et al. 2015; Catalán et al. 2018). The conclusions/limitations of these studies should be added/discussed in the introduction in order to clearly identify the big picture of the manuscript.

The last paragraph is I think the most interesting part as the authors propose a new model based on the nutrient spiralling concept to quantify DOM/N/P interactions. However it is very technical (e.g. lines 76-85) and hard to follow for readers that are not

familiar with these concepts. Thus, while I fell that the INSBIRE is of potential interest, I was lost before the end of the introduction and didn't understand how it may differ from existing models. I think this paragraph should be reformulated in a more understandable way, details being provided in the Material and methods section.

Lines 45-47 : problem with references.

Line 48 : add references.

Line 60: add reference.

Lines 62-63: this sentence is quite abusive. Moreover, the authors face the same problem with their approach as they are not able to identify any transformation pathways (they only make hypothesis).

Methods:

Line 94: Please provide information regarding the water residence time of the study site.

Line 113: After how long the plateau was reached after leachate additions, and how it compares with the travelling time of the stream?

Line 138: some additions are very low compared to ambient DOC, so how the authors can be sure that they are measuring uptake for leachates and not from ambient DOM? Please provide more justification here.

Line 149: specify the number and origin of EEMs included in the PARAFAC model.

Line 181: overall the description of INSBIRE, including equation and hypothesis made, is very hard to follow. It seems that several choices are made but justification and/or implications on the model results are not provided. For instance, how is defined the threshold that determines if some data are removed or not (line 204-205)? How do the authors justify the addition of a product of power functions to include interaction, and what do they mean by interaction (line 239)? How do they determine if/when adding

the wetted width is beneficial (line 240) and how do they related wetted witch to stream surface bed and/or retention processes? I think that all the presentation of INSBIRE should be reconsidered. Also, I didn't see any figures about errors from the model.

Lines 197-199: I don't understand these sentences.

Lines 259-260: and? What does it imply?

Results & Discussion

Lines 306-307: some statistic tests would be helpful to measure the level of significance of trends.

Figure 4: this figure is confusing. What I see here is mixing between leachates and stream waters along the stream reach, while the authors argue that at point 0 the mixing is full. If I understand well, these data are data collected directly in the stream, it would be interesting also to see the data corrected for dilution. Table 5: hard to read.

Line 501 & 517: these statements are a lit bit ambitious.

References

Casas-Ruiz, J. P., N. Catalán, L. Gómez-Gener, and others. 2017. A tale of pipes and reactors: Controls on the in-stream dynamics of dissolved organic matter in rivers. Limnol. Oceanogr. 62: S85–S94. doi:10.1002/lno.10471

Catalán, N., J. P. Casas-Ruiz, M. I. Arce, and others. 2018. Behind the Scenes: Mechanisms Regulating Climatic Patterns of Dissolved Organic Carbon Uptake in Headwater Streams. Global Biogeochem. Cycles 32: 1528–1541. doi:10.1029/2018GB005919

Guillemette, F., and P. A. del Giorgio. 2012. Simultaneous consumption and production of fluorescent dissolved organic matter by lake bacterioplankton. Environ. Microbiol. 14: 1432–1443. doi:10.1111/j.1462-2920.2012.02728.x

Vonk, J. E., S. E. Tank, P. J. Mann, R. G. M. Spencer, C. C. Treat, R. G. Striegl, B. W. Abbott, and K. P. Wickland. 2015. Biodegradability of dissolved organic carbon in permafrost soils and aquatic systems: A meta-analysis. Biogeosciences 12: 6915–6930. doi:10.5194/bg-12-6915-2015

---

## Author Comment (AC1) · 23 Dec 2020

General comments:

We thank the reviewer for their constructive comments. We consider the provided ideas a valuable input to improve the manuscript and will revised the manuscript accordingly.

1. We fully understand the reviewer's concern about independence as this was also one of our major concerns in planning the experimental design. However, as the environment also changes naturally (e.g. discharge, temperature), different additions cannot be compared if the interval between them is too long. Thus, we have tried

to find a compromise for the length of the intervals between the different samplings based on our long-term experiences in nutrient additions experiments. Independence effects were reduced to a minimum through the following considerations and were also checked regularly during the entire experiment:

a) The added material did not induce unnatural concentrations in the stream, but created peaks equal to or below local rain events. As the additions were within the range of the natural variability of the stream, we do not expect any stimulation of biofilm growth through the additions. Biofilm samplings after each addition as well as between additions supported this assumption by showing no systematic change in enzymativ activities over the course of the experiment.

b) Additions were limited to a maximum of two times per week with an interval of at least 48 h between two consecutive samplings, allowing the system enough time to recover. We also observed no systematic change in uptake rates over the course of the experiment, supporting again the assumption that the additions did not stimulate biofilm communities and their metabolism. As we could not identify a stimulation from additional P-PO4 on the DOM uptake, we conclude that the additional P-PO4 had no significant impact on the metabolic processes. Regular water analyses also revealed that the system remained P-limited throughout the entire experiment.

c) Regarding the natural environmental changes, we were lucky to accomplish all our experiments during a period of stable weather conditions.

We will add the above mentioned information in the method section and also shortly discuss potential effects of repeated additions as suggested by the reviewer.

2. We understand the concern of the reviewer and will move detailed information on INSBIRE to the supplement material.

a. Thank you for this suggestion. We will keep the graphs as they can provide additional information (e.g. mixtures of distributions become visible through shoulders

in the curves). However, we will add the confidence intervals and the probability of overlaps.

3. We will take serious effort to improve the language and the structure of the manuscript (e.g. revising the language, paragraph and sentence structure, moving detailed information on the model to the supplement material and structuring the discussion according to the research questions).

a. We appreciate the comment of the reviewer and will revise the manuscript accordingly.

b. We appreciate the comment of the reviewer and will revise the manuscript accordingly.

Specific/Technical Comments:

Introduction

1. We will revise the manuscript accordingly and provide clear connections between ideas and topics addressed wherever necessary to improve the structure and help readers follow our concepts.

a. We will add more information on DOM composition as suggested.

2. We appreciate the comment of the reviewer and will elaborate a more suitable wording and improve the explanation of the concept.

Line 42: We will revise the sentence as suggested.

Line 43: We differentiate between streams and rivers more accurately.

Line 45-46: We will correct this.

Line 46-47: We agree that the toxicity of pesticides has no direct connection with our research and will remove it to avoid confusion.

Line 48: We will revise the sentence as suggested.

Line 61: will be changed.

Methods

1. Section 2.1: We will add the information.

2. Section 2.2: See comment above.

3. We will move most of section 2.7 to the supplement material and adjust the structure to show the formulas, variables and parameters in context with the research questions in a clear way.

a. The choice of a suitable functions is indeed a difficult part in the analysis and we agree with the reviewer, that the power function is ecologically questionable. Besides the power function, we tested a linear function, a Michaelis-Menten type function, an exponential function and an asymptotic regression function. With some background knowledge, all of those (and more) functions can be used within the INSBIRE approach. We decided to use the power function because those models showed the highest Bayes factors for most additions. The big advantage of the power function is that there is only one parameter to fit, which makes it less prone to over-fitting in complex models. Also, in our experiment, concentrations did not reach uptake limits. In such cases, uptake rate curves often exhibit a power function (such as e.g. the efficiency loss model described by numerous other authors), probably representing as the lower part of a saturation model within a concentration range below saturation and thus naturally met in the system.

b. As our stream was rather small, even small changes in discharge may create quite large differences in the wetted width and thus in the reactive surface area. In fact, the wetted width of our stream ranged between 2.6 and 7.2, so the difference is a factor of 2.8. Thus, wetted width is an important parameter to analyse differences between sampling dates not due to source effects and our results actually show different responses of the various DOM components to changes in wetted width.

We will use L for litres throughout.

Results 1. We can see the mentioned structural weaknesses and thank the reviewer for the suggestion. We will shift information accordingly during the revision.

2. We will follow the suggestion of the reviewer.

Discussion

1. We will follow the suggestion of the reviewer.

2. We will follow the suggestion of the reviewer and provide introductory sentences to the individual paragraphs/passages.

3. Correct, two components within the same degradation or production process are not interacting; thanks to the reviewer's comment, we have realized that the original phrasing of this statement was unfortunate and much too short to describe the full complexity of this idea. What we actually meant was: Mathematical interactions between two different DOM components given by the model may not necessarily come from real (ecological) interactions, such as in the SRP example described by the reviewer (where one component affects the uptake of the other component); they may originate from a degradation or production process of DOM, in which both components are involved at different stages, so that one component is an intermediate product of the other; unfortunately, we cannot check whether this assumption is true with the applied method of spectroscopic analyses; however, several other authors have found indications that especially DOM components, which increase during the DOM uptake, may actually be degradation products of other (decreasing) DOM components.

We will clarify the paragraph accordingly and explain and use exact terms for mathematical correlations (which do not imply any ecological meaning), ecological interactions, and dependent components occurring at different stages of the same process.

Because of the complexity of the topic, we think, that correlations might not be able to reveal subtle relations (that might be covered by hydrology e.g.) and therefore used a

multivariate regression.

Line 403-405: We will provide more context. What we meant was that previous studies have shown different uptake velocities for different DOM sources like our study, but the number of such studies is too low and the used leachates and studied systems too diverse to draw any general conclusions about the uptake of different natural and anthropogenic DOM sources (or component mixtures) yet.

Line 403-420: We appreciate the comment from the reviewer and will consider this during the revision.

Line 414: will be corrected

Line 421: will be changed to relationship throughout the text

Line 432: Yes, this should be Figure 6, thank you for the comment.

Section 4.2: From the reviewer's comments, we see a need in better expressing the differences between the uptake velocities of different fluorophores. We do not necessarily see a contradiction, because in some cases, the overlaps do not allow a distinct separation, while in others, there is a clear difference. Additionally, the uptakes of the fluorophores behave differently in relation to other compounds. We will add information on the actual differences and accordingly change the description.

---

## Author Comment (AC2) · 23 Dec 2020

General comment:

We thank the reviewer for their comments on the manuscript. From the comments, we see a need for text improvements and a shift in focus on the biogeochemical processes. Methodological descriptions will be properly described, linked to biogeochemical processes and our motivation and decisions will be explained accordingly. Details on INSBIRE that is only necessary for applications in other studies will be placed in the supplement material to allow a fluent reading for people interested in our findings on the carbon cycle in aquatic ecosystems.

[Figure]

However, we disagree with the reviewers comment that the used INSBIRE approach is done by "turning buttons to fit data without clear ideas about the processes behind". Quite on the contrary, all model assumptions are based on ecological models and processes, commonly known in aquatic biogeochemistry and observed in various other (nutrient) uptake studies, such as saturation uptake kinetics, nutrient spiralling, the nutrient efficiency loss model, hydrological retention, and uptake and transformation processes. We will describe these aspects more specifically.

The parametrization of the model is based on previous studies (mostly the nutrient spiralling concept, additionally the studies by Dodds and O'Brien, both were citet in many other studies). For nutrient addition studies these have proven suitable. INSBIRE extends these concepts from single to multiple compounds which needs to be proven useful, but can be one logical and justified next step from recent research. We explained the extension by the parameter l in lines 241 to 243 because we have not found something comparable in any other study. The initial introduction of l was solely motivated by knowledge about the processes and tested statistically in the following steps. We consider the Bayes factor a suitable metric in model selection to avoid over-parametrization and provided references to support this assumption.

INSBIRE quantifies relations in uptake processes by means of Bayesian posterior distributions but not by means of single values. This approach has proven beneficial in many ecological studies and is justified in lines 203-204 and 514-515. We will make sure, this idea is addressed earlier and more understandable in the manuscript.

We see difficulties in the interpretation because of the novelty of the approach and its application in only one study so far but our study (e.g figure 7) shows a clear picture of the trends in the data. Therefore mainly comparisons and understanding of the drivers are lacking so far. We plan to and see lots of potential to use INSBIRE on many other field and lab experiments where we measure concentrations at different times in the future to add to our understanding of the carbon cycle in aquatic ecosystems.

[Figure]

Introduction

The study aimed at analysing the in-stream uptake of different complex DOM sources and the relation of DOM uptake on the occurrence of and the interactions between different DOM fractions (including co-leached nutrients). The aim was not to study the interaction between nutrients and DOM uptake; thus, no nutrients were added, but some were naturally still contained in the organic matter leachates (albeit at generally low concentrations) and therefore also included in the analyses as parts of the DOM (similar to the organic DOM components). Thus, citing the studies about nutrient-DOC interactions on DOM uptake would, thus, give a completely wrong picture of the actual aim of our study. However, we will rephrase the introduction, clarify the aim, and especially point out the role of the nutrients as inorganic part of the DOM leachate to avoid confusion. We will also include the mentioned studies (and others) in the discussion to interpret interactions between the DOM uptake and the co-leached nutrients; unfortunately, studies in lake systems or lab incubations (Guillemette and el Giorgio, Vonk et al, and many others) or using artificial substrates such as acetate (Catalan et al) are only limited applicable for our study;

We want to stress that DOM addition studies, which look into the uptake and the interactions of the individual inorganic and organic components of the DOM, are scarce and the interpretations are complicated, which makes our study and the developed INSBIRE approach an important step towards unraveling the mechanisms of DOM bioavailability.

We will focus on usability and differences to existing models in the introduction and shift the technical details to the methods or supplement material.

Line 45-47: We will correct this

Line 48: We will add references

Line 60: We will add references

[Figure]

Line 62-63: We will rephrase this sentence. We wanted to stress that mass balance approaches work well with P and N, but are limited if dealing with C components. We do not claim to identify transformation pathways, although they would provide a much more detailed insight into DOM uptake, as the method of spectroscopic analyses is not suited.

Methods

Line 94: We will add mean water residence times for the stretches.

Line 113: We will add this information to the manuscript.

Line 138: Indeed, some additions were quite low mainly due to our attempt to keep peaks within a realistic range, methodological issues (restricted leaching from some sources, low pumping rate), and small environmental changes in background concentrations and discharge. We corrected plateau concentrations by background and removed measurements that deviated from the ambient concentration less then two times the measurement accuracy of our lab instruments to remove questionable values.

Line 149: We will add this information. The number of EEMs was 176 and their origin was the very stream, the experiment took place.

Line 181: We will add the following information: We removed measurements which deviated from the ambient concentration less then two times the measurement accuracy of our lab instruments. The power function was chosen after testing several functions found in various nutrient addition studies (power function, linear function, Michaelis-Menten type function, exponential function and asymptotic regression function) and calculating the Bayes factor of the models. In most of the cases, the power function showed the best fit. Besides, the power function has only one parameter and is therefore less prone to over-fitting. This fact and the better comparability amongst the relations was the reason to use the power function throughout the study. We are

aware, that the power function does not provide an upper limit of the uptake, which would be ecologically sound. However, as the power function has been described and used in various other uptake studies, where saturation was not reached, we consider this a valid decision. The wetted width (as all other influencing factors) was added in cases, where the Bayes factor supported this decision. From a detectable influence of the wetted width, we concluded its importance for the respective processes. More wetted width means more surface covered with sediments and benthic microbes and we therefore think there is an important connection. The presented difference in impact of the wetted width is for us a clear hint, that some uptake processes might primarily take place at the sediment surface and others in the water column. However, we have realized from the two reviews that we need to better define and distinguish between the terms of mathematical interaction/correlations, influencing factors, and ecological interaction in the revised manuscript. Errors from the model are presented in figure S01.

Lines 197-199: The use and choice of prior distributions is an important and basic part of Bayesian statistics. We consider this explanation not within the scope of this study. We will add a reference.

Lines 259-260: In regression models of concentration data, it is common amongst statisticians to use lognormal error distributions as a first choice. Therefore it was our first choice as well, but our data did not fit this assumption and we used a normal error distribution instead. We mentioned this because we thought the question might arise. Exemplary studies, although from other geochemical fields of study: Ott (1990) A Physical Explanation of the Lognormality of Pollutant Concentrations, doi: 10.1080/10473289.1990.10466789, Ahrens (1954) The lognormal distribution of the elements (A fundamental law of geochemistry and its subsidiary), doi: 10.1016/0016-7037(54)90040-X.

Results & Discussion

Lines 306-307: We will add statistical information at that point.

Figure 4: We will improve the description and the readability of the graph. We ensured a lateral in-stream mixing at point 1 by measuring a uniform conductivity at several points in a cross section. Figure 4 shows indeed a decline of substances due to reactions. The graph can be changed to show the data corrected for dilution but would still show the same pattern. We will improve the readability of table 5.

Line 501 & 517: We will rephrase the statements. We only want to highlight the novelty and the potential of the approach as well as its limitations to enourage others to elaborate it, test and extend its applicability. We also wanted to highlight that the approach is versatile and based on well studied principles combined in a novel way.

References

References will be considered and added at the appropriate points.

---

## Author Response (AR1)

Dear Dr. Abril, dear reviewers,

thank you for your effort and your constructive reviews. These suggestions helped to significantly improve the manuscript in terms of clarity and compatibility with the readers of Biogeosciences. We changed and adapted large parts, which are described in detail in the point-wise answers to the reviewers' comments below.

Line numbers refer to the submitted manuscript with tracked changes.

**Reviewer 1**

*General Comments: The manuscript titled "Complex interactions of in-stream DOM and nutrient spiralling unravelled by Bayesian regression analysis" compares instream uptake of DOM differing in source/quality which has been relatively understudied in the literature. The authors pose an interesting research question well within the scope of biogeochemistry. A secondary objective of the manuscript was to refine a statistical model to estimate nutrient uptake that can provide estimates of uncertainty, account for nonlinearity, and allow for the addition of different nutrient fractionations (DOM optical properties here)to examine differences in uptake. The INSBIRE model appears robust and useful, but its mathematical/statistical evaluation is outside of my expertise. I appreciate the value in the author's research, but my opinion is that this manuscript needs major revisions before publication here or elsewhere.*

*1. My major concern is the lack of independence in the study design. Nutrient/DOC leachate additions were added to the same stream reach several times per week. Subsidies from these additions are going to stimulate periphyton communities increasing their metabolic activity and biomass. More metabolic activity/biomass of bacteria and algae is going to result in faster uptake velocities of DOC and nutrients (e.g., SRP). Thus, I do not think it is entirely fair to compare uptake rates of different leachates between multiple additions that occurred over a few weeks as periphyton communities would have had ample time to use these resource subsidies to increase production and potentially alter subsequent nutrient uptake measurements. Commentary is needed in the methods to justify the study design and in the discussion to address the potential effects of repeated resource subsidies on nutrient uptake rates.*

> We fully understand the reviewer's concern about independence as this was also our concern in planning the experimental design. However, as the environment also changes naturally (e.g. discharge, temperature), different additions cannot be compared if the interval between them is too long. Thus, we decided for a compromise concerning the length of the intervals between the different samplings based on our long-term experiences in nutrient additions experiments. Independence effects were reduced to a minimum through the following considerations and were also checked regularly during the entire experiment:
> a) The added material did not induce unnatural concentrations in the stream, but created peaks equal to or below local rain events. As the additions were within the range of the natural variability of the stream, we do not expect any stimulation of biofilm growth through the additions. Biofilm samplings after each addition as well as between additions supported this assumption by showing no systematic change in enzymatic activities over the course of the experiment.
> b) Additions were limited to a maximum of two times per week with an interval of at least 48 h between two consecutive samplings, allowing the system enough time to recover. We also observed no systematic change in uptake rates over the course of the experiment, supporting again the assumption that the additions did not stimulate biofilm communities and their metabolism. As we could not identify a stimulation from additional P-PO4 on the DOM uptake, we conclude that the additional P-PO4 had no significant impact on the

metabolic processes.

c) Regarding the natural environmental changes, we were lucky to accomplish all our experiments during a period of stable weather conditions.

We added the above mentioned information in the method section and also shortly discuss potential effects of repeated additions as suggested by the reviewer. (Lines 210-217)

*2. While I see the value in the INSBIRE model, I think there is too much commentary on it in the manuscript for this type of journal. I think these sections could be simplified to improve flow and keep the focus of the manuscript on DOM dynamics – as was outlined by the research questions in the introduction. Much of the detailed information could be added to a supplement for interested readers and anyone wanting to use the INSBIRE model in their own research.*

The section about the methodological approach to INSBIRE was mostly rewritten to make it easier to follow from a biogeochemical perspective (lines 302-464). We pronounce the differences to commonly applied analyses within the nutrient spiralling concept and explain the fundamentally important aspects in the manuscript. Most of the mathematical details (lines 353-427) were moved to section S1 in the supplement material.

*a. The use of probability distributions to describe uncertainty is a valuable aspect of Bayesian modelling and INSBIRE. However – while visually appealing – Figures 5and 6 are hard for the reader to interpret (e.g., how much do they overlap). The use of numerical 95% or 90% credible intervals instead of these distributions would be beneficial to the reader as the degree of overlap can be readily ascertained.*

Thank you for this suggestion. We have tried several options and decided to keep the graphs as they can provide complementing information (e.g. mixtures of distributions become visible through shoulders in the curves) but to add a table with Bayes factors of the probability (Lines 525ff, 535ff). By that, we can make the difference between uptake velocities easy to understand without overloading the graph. We consider this approach beneficial for the understanding.

*3. This manuscript would greatly benefit from a thorough editorial review to improve sentence structure, clarity, and flow. Some of the more complex sections in the methods, results, and discussion were hard to follow making it difficult to understand what was done and provide a comprehensive academic review of the manuscript. I have included some technical corrections below, but level of editing needed is beyond my capacity as a reviewer and my editorial comments are not complete.*

We improved the language and the structure of the manuscript significantly. Specifically, we revised the phrasing, paragraph and sentence structure, and moved more detailed information on the model to the supplement material. We furthermore structured the discussion more clearly according to the research questions. A native speaker rechecked the writing to increase the clarity of presentation.

*a. Paragraphs in the discussion and introduction would benefit from clear introduction and conclusion sentences to define the topic in the body. Clarity in structure would help the reader follow along.*

We appreciate the comment of the reviewer and revised the manuscript accordingly.

*b. Avoid starting sentences with "it", "these", "this", "they" ect. especially when multiple subjects are being referred to. Best to be specific and clear to help the reader follow along.*

We appreciate the comment of the reviewer and revised the manuscript accordingly.

***Specific/Technical Comments:***

***Introduction***

*1. The content of the introduction is good, but a thorough editorial review is needed to help the reader follow along and increase the connection between the content/concepts introduced.*

We revised the manuscript accordingly and provided clear connections between ideas and topics addressed wherever necessary to improve the structure and help readers follow our concepts.

*a. Could add a bit more information on DOM composition e.g., mixture of labile and recalcitrant compounds to the paragraph on lines 56-69.*

We added more information on DOM composition as suggested (lines 75-81).

*2. A better definition of dampening/stimulating effects is needed, and I do not think this is the most appropriate term. Nutrient uptake by stream communities has an upper limit simply due to scaled up enzyme kinetics. Once that uptake rate is reached, it will not increase any further even with increased nutrient concentrations. Your non-linear models are entirely appropriate, but the way that "dampening effects" are defined (line 80: "dampening effects of nutrient concentration on the uptake efficiency") and referred to throughout the manuscript is not clear to the reader. The lack of clarity here continues in the discussion in the paragraph on "dampening effects".*

We appreciate the comment of the reviewer. The term "dampening" was completely removed from the manuscript. Instead, we elaborated the "efficiency loss" model (decreased uptake with increasing concentration of the same component) as well as positive and negative interactions (decreased/increased uptake with increasing concentrations of another component) to clarify this issue.

*Line 42: sentence is vague and could likely be connected to the following sentence to add context*

This was solved by rewriting the introduction.

*Line 43: relate – is related to; stream and rivers are synonyms maybe differentiate with more context e.g., headwater streams, large rivers*

We differentiated between headwater streams and larger rivers accurately throughout the manuscript.

*Line 45-46: duplicate references*

Duplicate references were corrected.

*Line 46-47: DOM influences the toxicity of pesticides has no context and does not relate to your research question, consider removing – also duplicate references*

This sentence only lists a few of the various effects of DOM on important ecosystem processes, highlighting the key role of DOM for the aquatic ecosystem. Due to the

restructuring of the introduction, we think that mentioning this within a list of other effects will not confuse people.

*Line 48: "this capacity" – what capacity? Can link to the previous sentence to add context*

The misunderstanding was solved by rewriting the introduction.

*Line 61: is produced – are produced*

This was solved by rewriting the introduction.

**Methods**

*1. Section 2.1 additional commentary – a sentence or two – on the Hydrological OpenAir Laboratory would be better than just a citation.*

We added the methodological approach and the aim of the HOAL. A link to the website was added for readers interested in past and current work. (lines 168-172)

*2. Section 2.2 see general comment on experimental design. Increased justification is needed.*

See comment general part 1.

*3. I think sections 2.6 and 2.7 need to be simplified and organized based on how you presented the research questions in the introduction to make the methods easier to follow. Two models were used to look at 1) random effects of DOM quality and 2)interactions? Clear definition of variables in the interaction models would be helpful.*

Section 2.6 and 2.7 were combined and fundamentally changed. Most of former section 2.7 was moved to the supplement material. We used the structure of the research questions to guide the reader through INSBIRE and provided necessary information, leaving the mathematical details for the interested reader in the supplements.

*a. For interactions did you only used the power function with your independent variables? Did you test different functions or just use the power one? I guess the choice of the power function was because it was used in previous studies. I think an exponential function or a logistic function, if you want to go one step further, is more ecologically appropriate.*

The choice of a suitable function is indeed a difficult part in the analysis and we agree with the reviewer, that the power function is ecologically questionable. Besides the power function, we tested a linear function, a Michaelis-Menten type function, an exponential function and an asymptotic regression function. With some background knowledge, all of those (and more) functions can be used within the INSBIRE approach. We decided to use the power function because those models showed the highest Bayes factors for most additions. The big advantage of the power function is that there is only one parameter to fit, which makes it less prone to over-fitting (discussed in e.g. McElreath, 2020, chapter 7). Also, in our experiment, concentrations hardly reached uptake limits. In such cases, uptake rate curves often exhibit a power function (such as e.g. the efficiency loss model described by numerous other authors), probably representing the lower part of a saturation model within a concentration range below saturation and thus naturally met in the system.

*b. I am confused by the addition of wetted width as an interaction. Sure, more surface area might equal more retention of DOM, but all measurements occurred in the same stream with little difference in discharge.*

As our stream was rather small, even small changes in discharge may create quite large differences in the wetted width and thus in the reactive surface area. In fact, the wetted width of our stream ranged between 2.6 and 7.2, so the difference is a factor of 2.8. Thus, wetted width is an important parameter to analyse differences between sampling dates not due to source effects and our results actually show different responses of the various DOM components to changes in wetted width.

*Throughout: units for liters should be L*

L was used for litres throughout the manuscript.

**Results**

*1. The result section had many method-like statements in it. Some of these statements provided information that was not present in the methods. A clear methods section that follows the research questions presented in the introduction would help to keep the results section as a summary of what was found.*

We clearly distinguished between methods and results and structured the revised manuscript accordingly (lines 467-468, 508-510, 553-555, 567-568).

*2. Results could be better structured around the 3 research questions from the introduction.*

We restructured the results section in alignment with the research questions.

**Discussion**

*1. Discussion could be better structured around the 3 research questions from the introduction*

We restructured the discussion section in alignment with the research questions.

*2. Stronger introduction sentences are needed set up what is going to be discussed in each paragraph.*

Large parts of the discussion section were heavily edited and we kept this suggestion in mind during the process.

*3. "Interactions among different DOM components, which indicate transformations of one substance into another during DOM processing" – an interaction means that the relationship between uptake and concentration of one component is dependent on the concentration of another component. For example, more SRP leads to a more rapid increase in the uptake of a component relative to concentration. The interaction models were difficult to understand, but the above conclusion does not make sense. If DOM was being broken down into different components, the uptake of one component would be positively associated with the concentration of a degradation product. Since all leachate additions differed in composition the concentration of individual components is confounded. Correlations among net uptake of the different components may be better suited to address this.*

Correct, two components within the same degradation or production process are not interacting; thanks to the reviewer's comment, we have realized that the original phrasing of this statement was unfortunate and much too short to describe the full complexity of this idea. What we actually meant was: Mathematical interactions between two different DOM components given by the model may not necessarily come from (ecological) interactions, such as in the SRP example described by the reviewer (where one component affects the uptake of the other component); they may originate from a degradation or production process of DOM, in which both components are involved at different stages, so that one component is an intermediate product of the other. Unfortunately, we cannot check whether this assumption is true with the applied method of spectroscopic analyses; however, several other authors have found indications that especially DOM components, which increase during the DOM uptake, may actually be degradation products of other (decreasing) DOM components.

Because of the complexity of the topic, we think, that correlations might not be able to reveal subtle relations (that might be covered by hydrology e.g.) and therefore used a multivariate regression approach.

We clarified passages of the manuscript accordingly and explained and used exact terms for mathematical interactions, ecological interactions, and dependent components occurring at different stages of the same process.

*Line 403-405: these two sentences contradict each other. Provide more context.*

What we meant was that previous studies have shown different uptake velocities for different DOM sources like our study, but the number of such studies is too low and the used leachates and studied systems too diverse to draw any general conclusions about the uptake of different natural and anthropogenic DOM sources (or component mixtures) yet. Through the rewriting of the discussion section, especially, the more precise analysis of the differences of $v_f$s, we could solve this contradiction.

*Line 403-420: This paragraph goes back and forth comparing uptake within your study and between your study and others. I suggest sticking with the latter and write a new paragraph for the between leachate differences you observed in your study. This new paragraph will also help transition into the next paragraph.*

The two topics were clearly separated.

*Line 414: dung allied – manure added*

This was solved by rewriting the discussion.

*Line 421: relation – relationship*

This was solved by a rewritten discussion.

*Line 432: should this be Figure 6?*

We corrected this cross reference.

*Section 4.2: first paragraph states that there was no major difference in DOM bioavailability indicated by the broad overlap of parameter ranges, but the second paragraph discusses differences in uptake of different DOM components. The two paragraphs contradict each other.*

From the reviewer's comments, we saw a need in better expressing the differences between the uptake velocities of different fluorophores. We solved the mentioned contradiction by quantifying the probability of a difference with the Bayes factor and a clearer description (lines 525ff, 535ff).

*Reviewer 2*

*In their manuscript entitled "Complex interactions of in-stream DOM and nutrient spiralling unravelled by Bayesian regression analysis", Pucher et al. investigate the inter-actions between DOM and nutrient uptakes in a small stream based on an experimental setup. They used five different leachates having contrasting DOM properties based on optical measurements (PARAFAC), added these leachates into the stream and then measured DOC and nutrient concentrations and DOM properties along a 215 m reach. For interpreting their data and calculate uptake velocities, they proposed a new approach based on the spiralling concept and called Interactions in Nutrient Spirals using BayesIan REgression (INSBIRE). The topic is of great interest, however the manuscript is hard to follow for readers that are not familiar with this type of approach. Furthermore, some clarification are required regarding the relevance of INSBIRE. Indeed, it seems to me that the model requires a lot of parametrization that is subjective (e.g. lines 365-378 or lines 390-400). It is like turning buttons to fit each data individually without clear ideas about the processes behind. It is therefore hard to interpret the data, as recognized by the authors (line 454 for instance) that finally can only make hypothesis on the processes occurring (e.g. lines 482-487). It doesn't seem to me that INSBIRE allows finally to investigate or quantify properly the interactions between DOM and nutrients uptakes, and I was also wondering to which extent this approach could be used by other researcher and/or in other study sites.*

We thank the reviewer for his/her comments on the manuscript. From the comments, we see a need for text improvements and a shift in focus on the biogeochemical processes. However, we disagree with the reviewer's comment that the used INSBIRE approach is done by "turning buttons to fit data without clear ideas about the processes behind". Quite on the contrary, all model assumptions are based on ecological models and processes, commonly known in aquatic biogeochemistry and observed in various other (nutrient) uptake studies, such as saturation uptake kinetics, nutrient spiralling, the nutrient efficiency loss model, hydrological retention, and uptake and transformation processes.

The parametrization of the model is based on previous studies (mostly the nutrient spiralling concept, additionally Dodds et al., 2002; O'Brien et al., 2007, all of them were cited in many other studies). For nutrient addition studies these have proven suitable. INSBIRE extends these concepts from single to multiple compounds which needs to be proven useful, but can be one logical and justified next step from recent research. We explained the extension by the parameter l in lines 345 to 348 because we have not found something comparable in any other study. The initial introduction of the additive parameter l to the model was solely motivated by knowledge about the processes and tested statistically in the following steps. We consider the Bayes factor a suitable metric in model selection to avoid over-parametrization and provided references to support this assumption.

INSBIRE quantifies relations in uptake processes by means of Bayesian posterior distributions but not by means of single values. This approach has proven beneficial in many ecological studies and is justified in lines 101-103 and 807-809. We addressed this idea more understandable in the manuscript.

We see difficulties in the interpretation because of the novelty of the approach and its application in only one study so far but our study (e.g. figure 7) shows clear trends in the data. We see a high potential in using INSBIRE in many other field and lab experiments, especially regarding interactions in multi-component mixtures..

Details on INSBIRE that are only necessary for applications in other studies were placed in

the supplement material to allow a fluent reading for people interested in our findings on the carbon cycle in aquatic ecosystems. We focused on a well-known theoretical, biogeochemical basis and a step-wise guidance to INSBIRE. Restructuring the methods section according to the research questions will help the reader to follow the intentions, decisions and aims.

**Introduction**

*The introduction lacks of context regarding the interactions between DOM and nutrients, and why this is an important issue. The two first paragraphs are very broad, focusing on the importance of DOM on the biogeochemical and ecological functioning of freshwater ecosystems and on the impact of agriculture on DOM (which is not the subject of study), and do not help the reader to understand why "the effects of changed DOM and nutrient supply on the DOM and nutrient uptake in streams remains in the dark" (lines 54-55) or why the authors "expect a complex interaction between the different DOM fractions and the available N and P to explicate the bioavailability and the aquatic retention of the DOM" (lines 66-67). The introduction could be improve by including more context about DOM/nutrient interactions based on previous works (e.g. Guillemette and del Giorgio 2012; Vonk et al. 2015; Catalán et al. 2018). The conclusions/limitations of these studies should be added/discussed in the introduction in order to clearly identify the big picture of the manuscript.*

The study aimed at analysing the in-stream uptake of different complex DOM sources and the relation of DOM uptake on the occurrence of and the interactions between different DOM fractions including co-leached nutrients. However, the aim was not to study the interaction between nutrients and DOM uptake; thus, no nutrients were added, but some were naturally still contained in the organic matter leachates (albeit at generally low concentrations) and therefore also included in the analyses as parts of the DOM leachates (similar to the organic DOM components). We decided against citing too many nutrient (and DOC) addition studies about nutrient-DOC interactions in the introduction not to mislead readers about the aims of our experiments (namely, that we tested the effects of external nutrient sources on the DOM uptake). For this reason, we have added "co-leached" to the term nutrients. We have also re-written the entire introduction, clarified the aims, and especially pointed out the role of the nutrients as inorganic part of the DOM leachate. We have also included the Catalan and Guillemette as well as other stuies in the introduction and discussion to interpret general interactions between the DOM uptake and (co-leached) nutrients. Unfortunately, studies in lake systems or lab incubations (Guillemette and el Giorgio, Vonk et al, and many others) or using artificial substrates such as acetate (Catalan et al) are only limited applicable for our study.
We want to stress that DOM addition studies, which investigate the uptake and the interactions of the individual inorganic and organic components of complex DOM sources, are scarce and the interpretations are complicated, which makes our study and the developed INSBIRE approach an important step towards unravelling the mechanisms of DOM bioavailability.

*The last paragraph is I think the most interesting part as the authors propose a new model based on the nutrient spiralling concept to quantify DOM/N/P interactions. How-ever it is very technical (e.g. lines 76-85) and hard to follow for readers that are not familiar with these concepts. Thus, while I fell that the INSBIRE is of potential inter-est, I was lost before the end of the introduction and didn't understand how it may differ from existing models. I think this paragraph should be reformulated in a more understandable way, details being provided in the Material and methods section.*

We focused on usability and differences to existing models in the introduction and shifted the technical details to the methods or supplement material, which improves readability.

*Lines 45-47: problem with references.*

References were corrected.

*Line 48: add references.*

We added references.

*Line 60: add reference.*

We added references.

*Lines 62-63: this sentence is quite abusive. Moreover, the authors face the same problem with their approach as they are not able to identify any transformation pathways(they only make hypothesis).*

We rephrased this part. We wanted to stress that mass balance approaches work well with P and N, but are limited if dealing with C components. We do not claim to identify transformation pathways, although they would provide a much more detailed insight into DOM uptake, as the method of spectroscopic analyses is not suitable for this purpose.

**Methods**

*Line 94: Please provide information regarding the water residence time of the study site.*

We added mean water residence times for the stretches (Figure 2).

*Line 113: After how long the plateau was reached after leachate additions, and how it compares with the travelling time of the stream?*

We added this information to the manuscript (Figure 2, lines 200-201).

*Line 138: some additions are very low compared to ambient DOC, so how the authors can be sure that they are measuring uptake for leachates and not from ambient DOM?Please provide more justification here.*

Indeed, some additions were quite low mainly due to our attempt to keep peaks within a realistic range, methodological issues (restricted leaching from some sources, low pumping rate), and small environmental changes in background concentrations and discharge. We corrected plateau concentrations by background and removed measurements that deviated from the ambient concentration less than two times the measurement accuracy of our lab instruments to remove questionable values. This information was added in lines 317-319.

*Line 149: specify the number and origin of EEMs included in the PARAFAC model.*

We added this information. The number of EEMs was 176 and their origin was the very stream, the experiment took place in (line 240).

*Line 181: overall the description of INSBIRE, including equation and hypothesis made, is very hard to follow. It seems that several choices are made but justification and/or implications on the model*

*results are not provided. For instance, how is defined the threshold that determines if some data are removed or not (line 204-205)? How do the authors justify the addition of a product of power functions to include interaction, and what do they mean by interaction (line 239)? How do they determine if/when adding the wetted width is beneficial (line 240) and how do they related wetted witch to stream surface bed and/or retention processes? I think that all the presentation of INSBIRE should be reconsidered. Also, I didn't see any figures about errors from the model.*

We added the following information: We removed measurements which deviated from the ambient concentration less than two times the measurement accuracy of our lab instruments (lines 317-319). The power function was chosen after testing several functions found in various nutrient addition studies (power function, linear function, Michaelis-Menten type function, exponential function and asymptotic regression function) and calculating the Bayes factor of the models. In most of the cases, the power function showed the best fit. Besides, the power function has only one parameter and is therefore less prone to over-fitting. This fact and the better comparability amongst the relations was the reason to use the power function throughout the study. We are aware, that the power function does not provide an upper limit of the uptake, which would be ecologically sound. However, as the power function has been described and used in various other uptake studies, where saturation was not reached, we consider this a valid decision (lines 333-341).

The wetted width (as all other influencing factors) was added in cases, where the Bayes factor supported this decision. From a detectable influence of the wetted width, we concluded its importance for the respective processes. More wetted width means more surface covered with sediments and benthic microbes and we, therefore, think there is an important connection. The presented difference in impact of the wetted width is for us a clear hint, that some uptake processes might primarily take place at the sediment surface and others in the water column. Previous studies have located retention processes either in the benthic zone or in the water column. We can show a quality/nutrient dependence and demonstrate a method for this inference (lines 349-352, 565ff).

However, we have realized from the two reviews that we need to better define and distinguish between the terms of mathematical interaction/correlations, influencing factors, and ecological interaction in the revised manuscript.

Errors from the model were presented in figure S01.

*Lines 197-199: I don't understand these sentences.*

We assume that you refer to the use of priors and agree that the use of priors is not explained in depth within the mentioned sentences. The use priors is an important and needed part of Bayesian statistics. However, due to the length of the manuscript we do not include a detailed explanation, but add a reference with an in-depth description (lines 307-309).

*Lines 259-260: and? What does it imply?*

Concentration data is often lognormal distributed, hence we first chose lognormal error distributions. Therefore, it was our first choice as well, but our data did not fit this assumption and we used a normal error distribution instead. We mentioned this because we thought the question might arise. Exemplary studies, although from other geochemical fields of study addressing error distributions are: Ott (1990) A Physical Explanation of the Lognormality of Pollutant Concentrations, doi: 10.1080/10473289.1990.10466789, Ahrens (1954) The lognormal distribution of the elements (A fundamental law of geochemistry and its subsidiary), doi: 10.1016/0016-7037(54)90040-X (Supplements, lines 63-64).

**Results & Discussion**

*Lines 306-307: some statistic tests would be helpful to measure the level of significance of trends.*

We calculated the Bayes factor in favour of an exponential decay over constant concentrations and could show the evidence for this assumption for all DOM fractions and SRP (lines 485-491).

*Figure 4: this figure is confusing. What I see here is mixing between leachates and stream waters along the stream reach, while the authors argue that at point 0 the mixing is full. If I understand well, these data are data collected directly in the stream,it would be interesting also to see the data corrected for dilution. Table 5: hard to read.*

We improved the description and the readability of the graph (Figure 4, line 493). Sampling letters were exchanged by the dates and separate x-axis ticks for background measurement dates were removed. We ensured a lateral in-stream mixing at point 1 by measuring a uniform conductivity at several points in a cross section. Figure 4 shows indeed a decline of substances due to reactions. The graph was changed to show the data corrected for dilution but still shows nearly the same pattern. Table 5 is now table 7 (lines 597ff). The contents of former table 5 were partly moved to the supplements to keep it simple within the manuscript.

*Line 501 & 517: these statements are a lit bit ambitious.*

We rephrased the statements. We only wanted to highlight the novelty and the potential of the approach as well as its limitations to encourage others to elaborate it, test and extend its applicability. We also wanted to highlight that the approach is versatile and based on well studied principles combined in a novel way.

**References**

*Casas-Ruiz, J. P., N. Catalán, L. Gómez-Gener, and others. 2017. A tale of pipesand reactors: Controls on the in-stream dynamics of dissolved organic matter in rivers.Limnol. Oceanogr. 62: S85–S94. doi:10.1002/lno.10471*

*Catalán, N., J. P. Casas-Ruiz, M. I. Arce, and others.2018.Behind theScenes: Mechanisms Regulating Climatic Patterns of Dissolved Organic CarbonUptake in Headwater Streams.Global Biogeochem.Cycles 32: 1528–1541.doi:10.1029/2018GB005919*

*Guillemette, F., and P. A. del Giorgio. 2012. Simultaneous consumption and productionof fluorescent dissolved organic matter by lake bacterioplankton. Environ. Microbiol.14: 1432–1443. doi:10.1111/j.1462-2920.2012.02728.x*

*Vonk, J. E., S. E. Tank, P. J. Mann, R. G. M. Spencer, C. C. Treat, R. G. Striegl, B.W. Abbott, and K. P. Wickland. 2015. Biodegradability of dissolved organic carbon in permafrost soils and aquatic systems: A meta-analysis. Biogeosciences 12: 6915–6930. doi:10.5194/bg-12-6915-2015*

Casas-Ruiz et al., Catalan et al. and Guillemette et al. were considered and added at the appropriate points.

---

## Author Response (AR2)

Dear Dr. Abril,

thank you, for your feedback on our manuscript. It was checked by a native speaker for grammar and spelling errors, redundancies overlooked after the extensive revision, and complicated phrases were corrected.
Besides, we merged the original results chapters 3.3.1 and 3.3.2 and deleted one level of headings there. Below, we answer all issues raised. The line numbers refer to the version with tracked changes.

Define DOC and SRP in the abstract
> *The abbreviations DOC and SRP were explained in the abstract (lines 18 and 29).*

L30 "while the others more or less resembled the bulk DOC uptake" > Others what?
> *We added "the other DOM components" (line 30).*

L46 "alter the toxicity" increase or decrease or both?
> *DOM can alter the toxicity in both directions, depending on the toxic substance. However, as this is of no relevance for our study, we deleted it (line 47).*

Section 2.1 Site description: it would informative to localize (as an additional zoomed panel in figure 1, maybe some photographs?) the "several inflows, two natural springs, six drainage pipes", the "site with groundwater infiltration" the "small wetland". The '(dense) grass growth on the banks", the "deciduous forest at the beginning and end of the study reach" as well as the location of stream discharge and water quality continuous monitoring, the "two lateral inflows", even if this information is resumed in Fig 2.
> *We did not include this information in our study map as it is of no relevance for the actual study. Our study reach did not include any inflows, as mentioned in L 138, as these would have interfered with our addition study. We fear that providing a lot of details about these inflows, may thus be misleading and confuse readers. A satellite image with the stream, the study reach and the inflows was added to the manuscript (line 130). Besides, a brochure with a detailed map of the study stream can be easily downloaded via the provided internet link in L 119. We have added this information there in brackets.*

Fig. 2 "downstream from 1 in m" do you mean downstream distance from point/station 1 in meters?
> *We made this clearer by using the circle symbol for point 1 instead of the letter 1 (line 148).*

L192 you use the term "terrain model" only once a time here in the mat and meth section. Dou you mean "digital terrain model"? Is this term necessary?
> *Thank you very much for this comment! We have corrected the manuscript and used the term "digital terrain model" (line 213). The reason for this is that this phrase refers to the generation of the terrain geometry. Subsequently, "model" or "1D model" is used in reference to the hydrodynamic calculation of the abiotic parameters.*

You jump from section 2.5 to 2.7 without section 2.6
> *Thank you for the comment, we corrected 2.7 to 2.6 (line 221).*

L219 what do you mean by "while vf should compensate this problem", what problem? Why is it a problem? Please rephrase/explain
> *Thank you, for pointing this out. We changed to "while vf is independent of discharge" (line 244, see also Dodds et al 2002; Stream Solute Workshop 1990, and many others).*

L220 "U incorporates the concentration of the solvent", not clear, please better explain
> *We rephrased the sentence and made this more explicit (line 245).*

L222"because the compensation of hydrologic conditions makes general uptake patterns better visible" not sure what you mean here do you mean that Vf is compensating the changes in

discharge, making the changes in other parameters and particularly the biological ones, better reflected by Vf than U or Sw?

*Yes, correct. As changes in discharge are compensated, vf reflects biogeochemical processes better. We have rephrased the statement accordingly (lines 247 ff).*

L228 > "by doing that"

*We corrected the text accordingly (line 255).*

L252 "these models" What models? why plural?

*Fitting Eq. 1 with data from one DOM component or nutrient is considered to be one model. Because we investigated several DOM components and nutrients we dealt with many models. This sentence was rephrased: We decided to present only the results of the power function (Eq. 1) because its inclusion in the models for the different DOM components showed the highest BFs (highest probability to explain the observed data) in most cases (lines 285 ff).*

L254&256 concentrations of what?

*We clarified, what concentrations was meant. ...the concentrations of the DOM components and the co-leached nutrients … (lines 288-289).*

L260 thus "we" did not; "restrict the sign" do you mit "fix" or "impose" the sign?

*The sentence was changed to "...and thus we did not constrain the sign of mi." We think, this is a well suiting term that makes it clear, what we did (line 294).*

L262 "this is significant" what is significant?

*The sentence was rephrased and split up to "However, such a total collapse is not expected for DOM fractions since microbes can use other C sources. Thus, we incorporated an added …..…" (lines 296 ff)*

L281&289 not "by that" > "by doing that" (I doubt the paper has been edited by a native speaker)

*Thank you, for addressing this! The sentences were changed accordingly (line 334).*

L299&300 not sure "resemblance" & "resembled" are appropriate words. Similarity? Similar…

*Yes, the word "resemble" is used correctly here as confirmed by a native speaker. However, we changed the first sentence to "which showed a similar fluorescence as pure quinone" to be more precise (line 352).*

L300: "composition" if fluorescence is the criteria, not sure "composition" the appropriate word

*This is a common term in this context (as well as DOM quality). Please see e.g. Stedmon et al. 2005, Fasching et al 2020, Stedmon et al. 2008, Graeber et al 2012, and many others (line 354).*

L331: "these three" what? Revise phrasing

*We added "leachates" to make this clear (line 384).*

L333-334 where can we see the lower DOC peaks?

*This can be seen in Figure 4. We added a cross-reference in the manuscript (line 385).*

L335 "This demonstrates how a low number of observations or erroneous data influences results in Bayesian statistics" how do they influence? Please be more explicit

*They lead to a broader posterior probability density. We have added this information (line 386).*

L336 "However, although we cannot make reliable statements in all relation to other leachates, we get the probable range of uptake velocities" what statement? What relation? Please explain/rephrase

*We have merged the upper part with this statement (and shortened it) for better readability: "During nettles and pig dung leachate additions, the DOC peaks were lower (Fig. 4) and*

*measurement errors had a higher influence, leading to broader posterior probability densities (Figure 5). This hampered a clear separation of the vf of nettle and pig dung leachates from the vf probabilities of the other leachates. Nevertheless, we can assume that ...…" (lines 385 ff)*

L345 taken up slower, WHEREAS SRP,… language editing is needed
> *We apologize. These error happened during the extensive revision and was obviously overlooked. We corrected the sentence accordingly (line 407).*

L373 "Although the model improved decisively in comparison to the one without interactions" improve language
> *The sentence was changed to: "While the Tyr (C6) model including the interaction with Hum-ter (C2) improved decisively compared to the simple model without interaction terms, the best performance could still be reached with the Tyr (C6) model including the sampling date." (lines 440 ff)*

L376 delete "no additional information could be gained from the available data." Write "For Hum-micter (C3) and N-NO3, we found no effects of variable collinearity within the models"
> *Thank you, for pointing this out. The sentence was changed to "For Hum-micter (C3) and N-NO3, no effects of other variables could be identified with our models." (lines 444 ff)*

L380 delete "we found substantial evidence that"
> *This part of the sentence was deleted.*

L381 Not sure what is the meaning of "concurred" here
> *We meant "occurred simultaneously" and rephrased this statement (lines 450 ff).*

L392 "to get the effect, their uncertainty would have on the model" > to get the effect of their uncertainty on the model results?
> *The sentence was changed accordingly (lines 461 ff).*

L396 "Hence, we do not expect a more sophisticated model to reveal any more details" not sure what you mean here
> *If the measurement errors and the model residuals are close, the information that can be drawn from the data is exhausted. We added this information to the manuscript (lines 465 ff).*

L398 "The higher error of the model compared to the assumed effect of the measurements on the accuracy shows that" please improve phraseology.
> *The sentence was changed to "Since the model residuals are higher than the assumed effect of the measurements the model has still potential for improvement…" (lines 468 ff).*

L404 "as also observed"
> *Thank you. The sentence was changed accordingly (line 487).*

L411 "Interestingly, the same sequence of increasing uptake velocities from cow dung leachate to leaf leachate and corn leachate was observed in a laboratory flume experiment using the same organic matter sources as this field study, but different sediments (Weigelhofer et al., 2020)"
> *The sentence was changed accordingly (lines 493 ff).*

L414 "There, however..." improve language
> *Thank you for pointing this out. The sentence was changed accordingly (line 496).*

L418-L419 So what? Do you mean that "in addition, leaf leachate uptake also varies with P fertilization of trees"?

*More or less; long-term phosphorus fertilization of trees led to P-enriched leaf leachates which showed a higher uptake than the leaves from non-fertilized trees; we have rephrased this part (lines 500 ff).*

L420 I would say it lies in the upper range
*In this case not, if one considers the distribution of vfs found in the literature; in this case, our vf lies close to the median (line 505).*

L427 "was taken up slowest" > most slowly. It looks like this MS has NOT been edited by a native speaker as the authors say
*This has either been overlooked by the native speaker or wrongly been accepted after the quite substantial revision. We apologize. The sentence was changed to "… showed the slowest vf" (line 512).*

L430 remove "about"
*"About" was removed (line 516).*

L438 "was only intermediate in our study" not sure what you mean here
*We meant lying between the fast and the slow components in our study, although fast uptake is assumed according to the literature. We rephrased it to: "showed only medium uptake velocities" (line 518).*

L437 "The uptake of N-NO3 was the lowest of all components due to its high background concentrations in the water column" not clear why high background NO3 induce lower NO3 uptake, looks like counterintuitive
*Because N is in excess, and thus microbial N uptake is strongly P or C limited; besides, if present, NH4 is preferentially taken up by bacteria due to its lower molecular weight, resulting in a further decrease of NO3 uptake and in nitrate behaving like a conservative tracer rather than a reactive nutrient; this has been shown by numerous nutrient addition studies, such as Dodds et al. (2014, https://doi.org/10.1007/s100210000050), Wymore et al. (2016, https://doi.org/10.1002/2016GB005468), Dodds et al. (2003, 10.1007/s00442-004-1599-y)*

L443 "huge number of different bacterial strains" not sure what you mean here
*"Strain" is the scientific term in aquatic microbiology for "bacterial species" detected by molecular analyses; it means that most of present bacterial species showed this ratio (see e.g. Godwin et al.)*

L445 "we do not believe that... Rather, we assume that". Based on what facts? Looks speculative
*Our assumption is based on the following: Because C:P ratio were already in an optimal range during background conditions and even decreased during the additions, pointing rather to C-limitation than P-limitation in our stream (see statements immediately above); and because SRP and DOC uptake were not correlated (see e.g. Tab 6 and results) and N was not limiting; if stoichiometry would have controlled P uptake in our study, we should have seen a clear link between P uptake and DOC concentrations and much higher C:P ratios; for references, see, e.g., by Cross et al, 2005 and Stutter et al. 2020. We have added a short explanation about the P and C demand of auto- and heterotrophs (lines 530 ff).*

L458 "However, as the molar ratios of C:P were low in our stream, showing no P limitation, and we also did not raise the SRP concentrations in our stream additionally to the P content of the leachates, SRP-related effects on DOC retention might have stayed uncovered" Almost impossible to understand. Please improve phraseology.
*We split the statement into 3 sentences: "However, the molar C:P ratios were low in our stream, showing no P limitation. Besides, most P peaks during the additions were rather*

*small, containing only the leached P from the DOM sources. Thus, potentially stimulating effects of SRP on the DOC retention may have remained undetected. (lines 546 ff)*

Same for the next sentence "performed much better" performed what? You mean have better performance?

*The term was changed to "showed a higher probability of explaining the measured values" (lines 551 and 552).*

L466 and on several other occasions "This efficiency loss can be explained by the processing capacity of the stream ecosystem, which is influenced by adaptions of the microbial community to usually occurring concentrations" this is a very vague explanation. Usually, biological uptake increases with the concentration of the substrate (first order or Michaelis – Menten kinetics) what is happening here?

*Michaelis-Menten is not the only model for uptake kinetics in streams; more often, scientists have observed a Power function that was termed "Efficiency loss model" (see, e.g., O'Brien et al., 2007; Mulholland et al., 2009; Merseburger et al., 2011); here, uptake increases with increasing concentrations, but with a k < 1; this happens especially in agricultural streams where the community is adapted to chronic loading (lower uptake rates, but more flexibility towards higher loadings, thus delayed saturation). We have rephrased the statement and added more information to improve the readability and intelligibility also for readers not familiar with addition studies (lines 557 ff).*

L467: how can transport become limiting if availability of the substrate increases? This looks strange. I can see that some detailed explanations are given later, I think the discussion should be better worded and structured in order to avoid such crude counter-intuitive statements at the beginning of a paragraph.

*Availability of substrate and transport limitations are two key factors influencing uptake, but are not necessarily related (or rather, one of these factors usually dominates the uptake, while the other is neglectable; transport limitation (e.g. due to clogged sediments or thick biofilms) can prevent the reactive solvent to reach the reactive site, thus limiting uptake despite a high availability of the substrate. We have rephrased the statement together with the efficiency loss mentioned above (lines 561 ff).*

Page 23 and 24 and at many places in the MS: no need to separate the text in so many small paragraphs

*We have merged paragraphs and also chapters of the results section*

L488: please explicit what are the "substantial evidences". Or write later "these evidences are:… good degradation conditions, ideal stoichiometric ratios", etc… you writing style is too imprecise

*"Substantial evidence" is a common term when assessing probability based on BFs. We have made this more explicit in the methods section lines 204-206. Furthermore, we mention that the interpretation was based on BFs explicitly. "Good degradation conditions" are explained by the examples following (such as no transport limitation, ideal stoichiometric ratios, etc.); thus we do not see the "impreciseness" in this term; "ideal stoichiometric ratios" are quite commonly used in uptake studies (see e.g. Cross et al., 2005; Godwin and Cotner, 2018; Stutter et al., 2018, and many others) and are also explained in the introduction lines 51-55; again, we do not see large impreciseness here, but we have added "ideal stoichiometric C:N:P ratios of the organic source for the microbial metabolism" to make it clearer.*

L496: sorption on what? Sediments? Again, be precise. We often have to read 2-3 sentences before understanding the first one

*We have added "sediments or extracellular polymeric substances" (line 604).*

L508: "The INSBIRE approach was developed after the data from the experiment was acquired due to limitations in other data analysis methods developed for inorganic nutrient uptake (Stream Solute

Workshop, 1990), such as the lack of a strategy to handle interactions among DOM components." I did not understand

> *In our original manuscript, some readers expected the study to be primarily a method check of the model. This is not the case. The model was only developed after we already had the data (including the high natural variability in the field and thus not ideal to test the strength and weaknesses of the model). We wanted to clarify this here. For better intelligibility, we have deleted the part after "was acquired. Thus, our study represents a case study …" (lines 625 ff).*

I guess a detailed review by co-authors can improve the MS particularly section 4.4

> *All co-authors and a native speaker have read the revised manuscript. However, due to the extensive revision, the manuscript was difficult to read and check in change-track mode, and some errors have also been overlooked during the final combination of the different revised/controlled versions. We apologize. We have screened the manuscript closely and also it was also checked again by a second native speaker.*